

# Millennial-scale fluctuations of palaeo-ice margin at the southern fringe of the last Fennoscandian Ice Sheet

Karol Tylmann[1], Wojciech Wysota[2], Vincent Rinterknecht[3], Piotr Moska[4], Aleksandra Bielicka-Giełdoń[5], ASTER Team[*]

[1]Department of Geomorphology and Quaternary Geology, University of Gdańsk, Gdańsk, 80-308, Poland
[2]Department of Geology and Hydrogeology, Nicolaus Copernicus University, Toruń, 87-100, Poland
[3]Aix Marseille Univ, CNRS, IRD, INRAE, CEREGE, Aix-en-Provence, 13545, France
[4]Institute of Physics – Centre for Science and Education, Gliwice, 44-100, Poland
[5]Department of General and Inorganic Chemistry, University of Gdańsk, Gdańsk, 80-308, Poland
[*]A list of the team members appears at the end of the paper

*Correspondence to*: Karol Tylmann (k.tylmann@ug.edu.pl)

**Abstract.** The paper presents a study of reconstructing chronology and dynamics of palaeo-ice margin oscillations at the southern fringe of the last Fennoscandian Ice Sheet (FIS) based on combined luminescence and $^{10}$Be surface exposure dating. The study area is located in northern Poland close to the last FIS maximum limit. Luminescence method was used to date sandy deposits (fluvioglacial sediments and aeolian deposits filling fossil periglacial wedges) intercalating basal till layers, and the most likely age of the tills was constrained by Bayesian modelling. $^{10}$Be dating method was used on erratic boulders left during the final retreat of the last FIS and resting on at the surface of glacial landforms. Our results indicate millennial-scale oscillations of the last FIS in northern Poland between ~19 and ~17 ka. The last FIS retreated and re-advanced over a relatively short period of time (2–3 ka), leaving a lithostratigraphic record (basal tills) of three ice re-advances in a millennial cycle: $19.2 \pm 1.1$ ka, $17.8 \pm 0.5$ ka and $16.9 \pm 0.5$ ka. The paper presents the first terrestrial record of millennial-scale palaeo-ice margin oscillations at the southern fringe of the FIS during the last glacial cycle. We explore the dynamics of these oscillations and confront the proposed cycles of the southern FIS advances and retreats with existing patterns of the last deglaciation and millennial-scale fluctuations of the last FIS inferred from marine records.

## 1 Introduction

Ice sheets and glaciers are key components of a cryosphere coupled to climate, global sea level or ocean circulation (e.g., Clarke et al., 1999; Greve and Blatter, 2009; Fyke et al., 2018). Ice sheets fluctuations are good indicators of climate changes, as they tend to stay in equilibrium with regional climate, reacting on any long-term variations of temperature and precipitation by their mass balance adjustment. However, interactions between ice sheets and climate are complex. Cooling and warming affect expansion and shrinkage of glaciated areas, but on the other hand, the size of areas with permanent ice cover have significant impact on the climatic system, e.g. by controlling the magnitude of albedo, by delivering large amounts of cold freshwater into oceans or by diverting the jet stream circulation. Thus, ice sheets and glaciers are strongly



linked to climate, being in fact a key element in the ice-climatic system (e.g., Hahn et al., 2018; Noble et al., 2020). In the era of global warming, a lot of attention is given to understand their past and current trends and to feed models simulating

their future behaviour. The last two ice sheets, Antarctica and Greenland have been also monitored, and their shrinkage has been analysed in relation to changing climate (Thomas, 2001; Rignot et al., 2019). Our knowledge about interactions of the Pleistocene palaeo-ice sheets with past climate changes is however, much more limited, as glacial geological record is fragmentary and in many cases difficult to date (e.g., Fuchs and Owen, 2008; King et al., 2014; Davis, 2022). Therefore, in order to explore interactions between palaeo-ice sheets, such as the Fennoscandian or the Laurentide ice sheets, with

Pleistocene climate fluctuations it is essentially important to link available geological record with timing of palaeo-ice sheets advances and retreats. It enables to correlate these spatial fluctuations with palaeoclimatic records available from ice cores, marine sediments, loess sequences or other lacustrine archives for example (Levy et al., 2018; Rea et al., 2018; Nawrocki et al., 2019). Dating terrestrial glacial record is however challenging, mainly due to the very dynamic nature of glacial environment, resulting in great lateral and vertical variations of sediments, presence of erosional gaps and deformations as

well as post-depositional reworking (Brodzikowski and van Loon, 1987; Kurjański et al., 2020).

Here we present a study reconstructing the chronology and dynamics of palaeo-ice margin oscillations based on combined luminescence and $^{10}$Be surface exposure dating. The study was conducted in northern Poland, at the southern fringe of the last Fennoscandian Ice Sheet (FIS). Luminescence method was used to date sandy deposits intercalating basal till layers and the most likely age of the tills was constrained by Bayesian modeling. $^{10}$Be method was used to date erratic boulders left by

the last FIS and resting on a surface of conspicuous glacial landforms. The paper presents the first terrestrial record of millennial-scale palaeo-ice margin oscillations at the southern fringe of the FIS during the last glacial cycle. We explore the dynamics of these oscillations, and we confront proposed cycles of the southern FIS advances and retreats with existing pattern of the last deglaciation and millennial-scale fluctuations of the last FIS inferred from marine records.

## 2 Study area and dating sites

### 2.1 Location

The study area is located in northern Poland in the region covered by the last FIS, in the very close vicinity of its maximum limit (Fig. 1a). It covers the region of a fresh glacial landscape shaped in the Late Pleistocene during the last ice sheet advance and retreat, which in this part of the northern Polish Lowland occurred around 22–18 ka BP (Tylmann et al., 2019; Hughes et al., 2022; Marks et al., 2022). The area is located within elevated morainic upland (the Lubawa Upland), where

the highest elevations exceed 300 m a.s.l., and are located up to 200 m higher than surrounding lowlands and valleys (Fig. 1b). Topography of this region is highly diversified with conspicuous moraine hillocks and deeply incised valleys creating local denivelations up to 50–70 m. Variability of such fresh glacial relief is a result of glaciotectonic deformations repeatedly occurring during several Pleistocene glaciations in this region (Marks, 1979; Gałązka et al., 2009) and intensive meltwater erosion of the ice bed and ice sheet`s foreland during the last deglaciation (Tylmann, 2014).





Sediments outcrop where luminescence dating was conducted is located on the north-western slope of the Lubawa Upland, within one of the moraine hillocks which occur in this area (Figs. 1b and 2b). The site is a gravel pit in Rożental, where the sequence of up to 10 m thick Pleistocene glacial deposits is exposed (Fig. 3). The origin of this sedimentary sequence was described in details elsewhere (Tylmann and Wysota, 2011; Tylmann et al., 2014). Here, we focus on a brief description of the main sedimentary units and luminescence dating of glaciofluvial deposits and fossil periglacial sand wedges. [10]Be dating

was applied to large erratic boulders resting on the glacial landforms. Location of dated boulders is presented in Fig. 1.

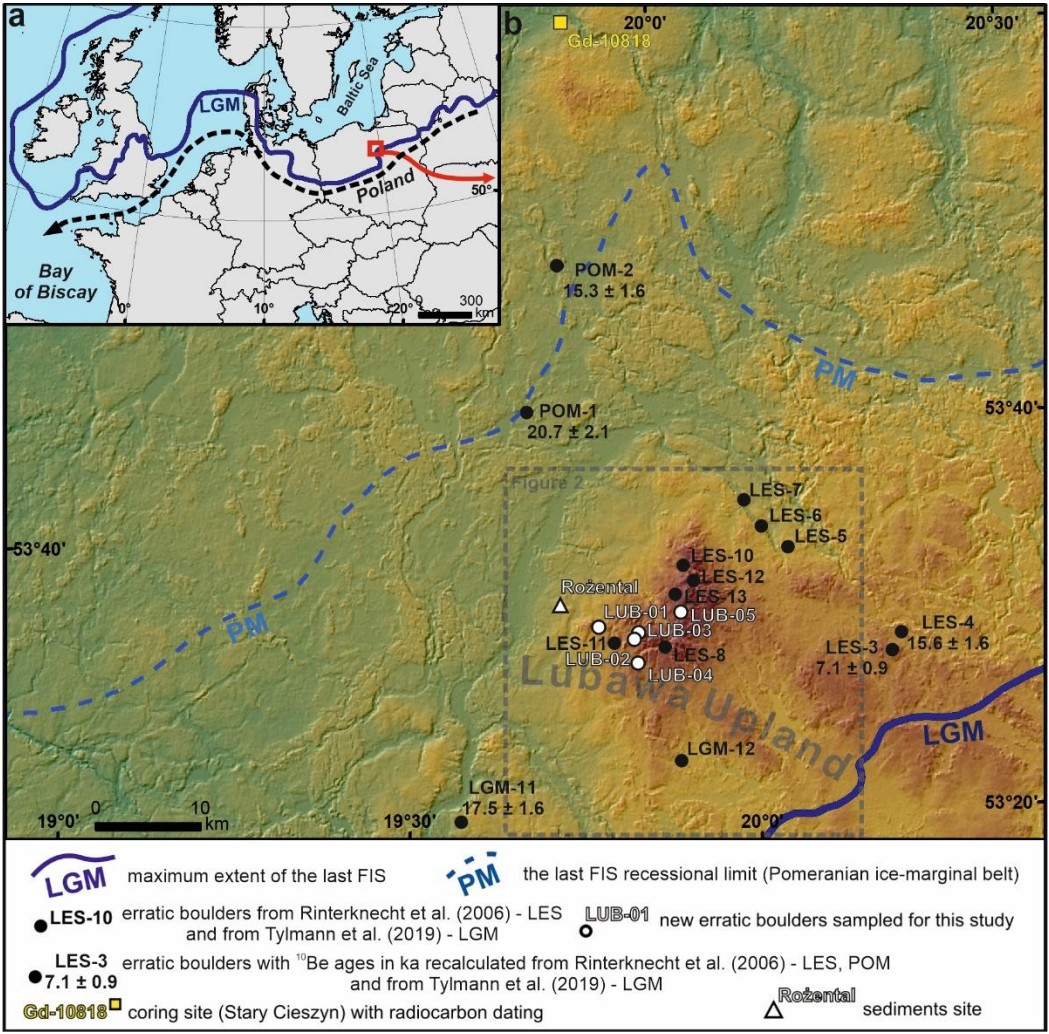

**Figure 1:** (a) Study area against the maximum extent of the last FIS in Europe. The Channel River route during the Last Glacial Maximum (LGM) is marked with black dashed line (Toucanne et al., 2015). (b) Digital elevation model (SRTM) of northern Poland with location of sediments site (Rożental), large erratic boulders (LES-3 – 13, LGM-11 – 12, LUB-01 – 05 and POM-1 – 2) and coring site where
radiocarbon dating has been done (Niewiarowski, 2003).



## 2.2 Glacial record

### 2.2.1 Landforms

The most characteristic elements of the glacial relief in the study area are numerous ridges of a well-preserved terminal moraines and deeply incised valleys of various origin (Fig. 2a). Elevation of terminal moraine ridges ranges from about 160
m a.s.l. to above 300 m a.s.l., and the elevations of valley floors is between 90 m a.s.l. and about 280 m a.s.l. Terminal moraines occur mainly in the central, relatively highly elevated part of the Lubawa Upland as well as on its western and eastern slopes. On the western slope of the Lubawa Upland most of terminal moraines are oriented NE-SW, while on the eastern slope moraines orientation is much more diversified (Fig. 2a). Most of the terminal moraines have an asymmetric morphological cross-profile.

Valley systems are also clearly visible in the topography of the Lubawa Upland (especially in its relatively highly elevated central part), and it consists of three types of glacial valleys: subglacial, ice-marginal and proglacial valleys (*sensu* Greenwood et al., 2007). Subglacial valleys are almost entirely oriented NW-SE, and most of them are currently occupied by rivers or lakes. They have undulating longitudinal profiles and some of them cut elevated morainic areas – having convex longitudinal profiles. Spatial distribution of these valleys indicates that they are mostly perpendicular or oblique to the
terminal moraine ridges (Fig. 2a). The largest landforms have complex morphology and they may be classified as subglacial tunnel valleys, while others with simpler morphology are probably subglacial channels (*sensu* Clayton et al, 1999). Ice-marginal valleys are oriented NE-SW and they occur mostly on western and north-western slopes of the Lubawa Upland. These valleys are mostly parallel or oblique to the terminal moraine ridges, and they are perpendicular or oblique to subglacial valleys. On the western slope of the Lubawa Upland ice-marginal valleys occur in a 'step-like' morphological
sequence with parallel valleys running along the slope (Fig. 2a). A few valleys which might be classified as routes of former proglacial meltwater outflow can be found on the southern and south-eastern slope of the highest elevated central part of the Lubawa Upland (Fig. 2a). Proglacial valleys are oriented N-S and NW-SE, and they run downslope towards the outwash plains which occur in the southern and south-eastern parts of the study area.

### 2.2.2 Sediments

Glacial till and fluvioglacial/fluvial sand and gravel dominate in the surface lithology of the Lubawa Upland. Till and related deposits (unsorted "dirty" gravels with boulders) are associated with moraine plateaux and terminal moraines, which occur mainly in south-western, western and north-western parts, in the elevated central part, as well as in south-eastern and eastern parts of the study area. Fluvioglacial sand and gravel are associated with outwash plains, which occur mostly on southern, south-eastern and eastern slopes of the Lubawa Upland, as well as in association with terraces within wide ice-marginal
valleys found in the north-western corner of the study area and with large subglacial valleys (Fig. 2b). Outwash plains in south-eastern and eastern parts of the study area are usually narrow, elongated tracks of glacial meltwater runoff, located in between the higher moraine uplands and oriented NW-SE. Besides glacial till and fluvioglacial/fluvial sand and gravel,



glaciolacustrine and lacustrine silt and clay also occur, mainly as a few isolated patches in south-western and southern part of the region. The spatial distribution of most of surface sediments is a result of the last FIS dynamics in the Lubawa Upland

and the process of its deglaciation (Tylmann, 2014). This region is rich in massive erratic boulders and boulder fields, which commonly occur at the surface of moraine plateau, moraine hillocks or within the glacial valleys. The largest of them were the subject of [10]Be dating (this study; Rinterknecht et al., 2005, 2006; Tylmann et al., 2019) (Fig. 2b). Fluvial sand and gravel are associated with river channels, while valleys and lake basins are filled with Late Glacial and Holocene peat and deluvium (Fig. 2b).

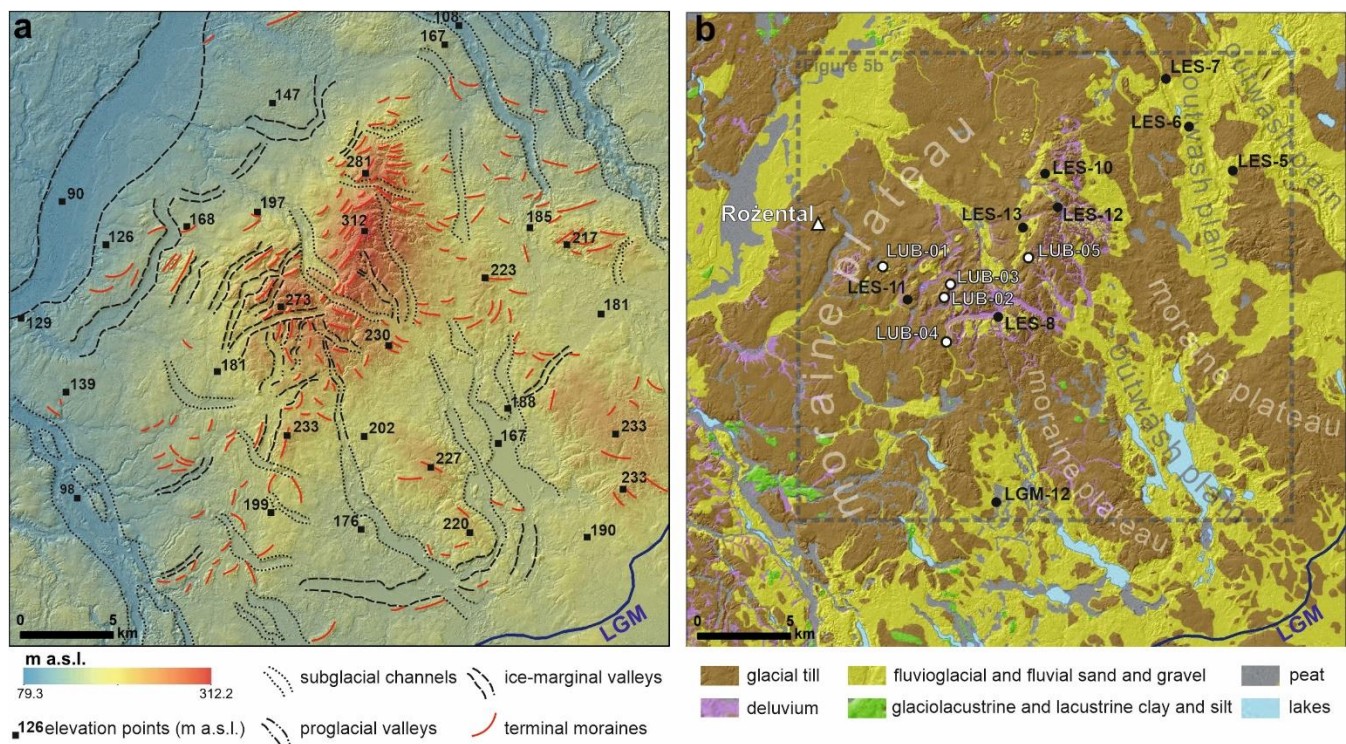


**Figure 2:** (a) High-resolution (1 m) digital elevation model (LiDAR) of the study area with the main glacial landforms. (b) Surface sediments of the study area draped over the digital terrain model. Distribution of surface sediments was compiled based on Detailed Geological Map of Poland (Gałązka and Marks, 1997; Gałązka, 2003, 2006, 2009; Wełniak, 2002). The main moraine plateaux and outwash plains are indicated as well as locations of the Rożental site and sampled erratic boulders.

Sequence of Pleistocene glacial deposits exposed at the gravel pit in Rożental consists of fluvioglacial sand and gravel covered by glacial till layers (Fig. 3a). Fluvioglacial unit (Rz1) is dominated by medium- to large-scale sandy-gravelly and gravelly-sandy beds with horizontal stratification. Most of these beds reveals normal grading and contacts between particular lithofacies are erosional. Occasionally, sand beds and lithofacies with through-cross bedding also occur. Within sandy and sandy-gravelly beds oversized clasts are very common. Fluvioglacial unit Rz1 is covered by a 2.5 m thick massive till (unit

Rz2) with a fossil periglacial structures (sand wedges) occurring in two separate horizons – K1 and K2 (Fig. 3b). This indicates that unit Rz2 consists of three separate till subunits: Rz2a, Rz2b and Rz2c. Distinct features of tills such as: (1)





sharp, planar contact with underlying deposits (Fig. 3c), (2) embedded clasts with flat upper surface and ploughing marks (Fig. 3d) and (3) glacial striations on clast surface (Fig. 3e), indicate that these are subglacial traction tills – a lithostratigraphic record of at least three ice advances and retreats postdating sedimentation of the fluvioglacial unit Rz1

(Fig. 3b; Tylmann, 2014; Tylmann et al., 2014).

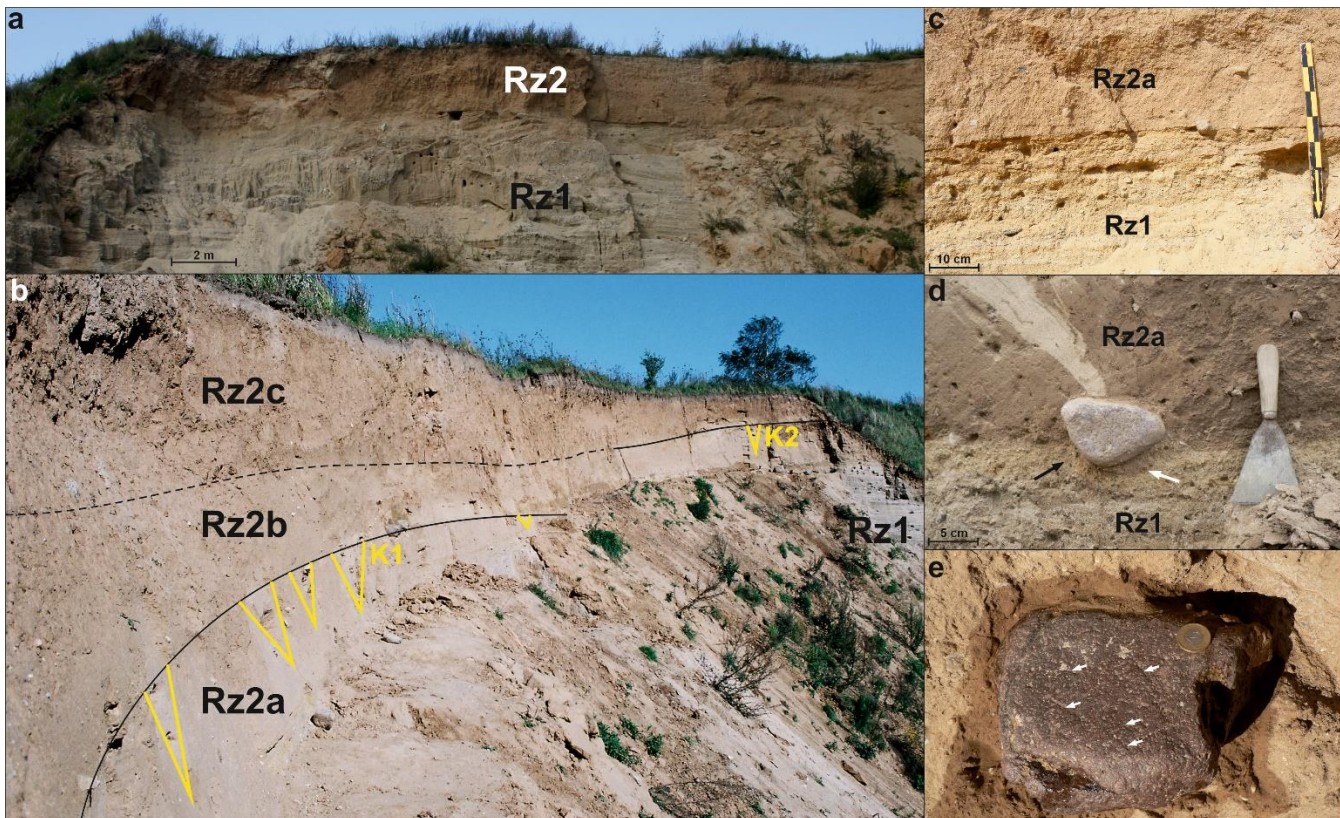

**Figure 3:** Sequence of the Pleistocene deposits exposed at Rożental site. (a) Panoramic picture of the sequence with two main sedimentary units indicated: fluvioglacial sand and gravel (unit Rz1) and basal till (unit Rz2). (b) Upper part of the sequence with periglacial horizons K1 and K2, and till subunits Rz2a, Rz2b and Rz2c. (c) Sharp, planar contact between units Rz1 and Rz2a. (d) Clast with flattened upper
surface embedded at the contact between units Rz1 and Rz2a. Ploughing mark (black arrow) and dimictic injection (white arrow) are visible below the clast. (e) Flattened upper surface of clast embedded at the bottom of unit Rz2b; glacial striations are marked with white arrows.

## 3. Methods

### 3.1 Luminescence dating and Bayesian analysis

Samples for OSL dating were taken from sandy beds of fluvioglacial unit Rz1 (three samples) and from aeolian sand filling fossil periglacial wedges of horizons K1 and K2 (eight samples). Sediments were sampled with plastic tubes pressed into the vertical section of deposits and secured with black PCV tape in order to protect samples from sunlight (Fig. 4a). Samples were analysed at the Gliwice Luminescence Laboratory (Moska et al., 2021) and only material taken from the middle parts





of the plastic tubes was processed. For OSL measurements, grains of quartz (45–63 μm) were extracted from the sediments
by routine treatment with 20% hydrochloric acid (HCl) and 20% hydrogen peroxide ($H_2O_2$) to remove carbonates and
organic matter formed in the samples. The final step of preparation was treatment with concentrated (40%) hydrofluoric acid
(HF) for 40 minutes to remove remaining of other minerals and outer layer of quartz (~10 μm, responsible for absorbing the
alpha radiation; Aitken, 1985). All OSL measurements were performed using an automated Daybreak 2200 TL/OSL reader
(Bortolot, 2000) fitted with a calibrated $^{90}Sr/^{90}Y$ beta source delivering about 2.7 Gy/min to grains at sample position.
Daybreak 2200 uses blue diodes (470 ± 4 nm) delivering about 60 mW/cm$^2$ at the sample position after passing through
BG39 filters. Equivalent doses were determined using the single-aliquot regenerative-dose (SAR) protocol (Murray and
Wintle, 2000). The SAR dose response curves were best represented by a single saturating exponential function. Final
equivalent dose ($D_e$) values were calculated using the Minimum Age Model (MAM) or Central Age Model (CAM)
(Galbraith et al., 1999). To determine the most adequate statistical model for equivalent dose calculation the overdispersion
parameter (σOD) was calculated using the R package 'Luminescence' (Kreutzer et al., 2012). We applied the CAM model in
calculations when σOD was below 20%, while the MAM model was applied when σOD did not meet this criterion. In order
to assess the dose rates ($D_r$) that arise from decay chains and potassium we used high-resolution Canberra gamma
spectrometry, calibrated with the reference materials, namely IAEA-RGU-1, IAEARGTh-1, and IAEA-RGK-1 obtained
from International Atomic Energy Agency reference materials. The dry dose rates (Guerin et al., 2011) were adjusted for
water content, following Aitken (1985). The cosmic ray dose-rate to the site follows the calculations suggested by Prescott
and Stephan (1982). The calculated OSL ages are reported in ka with 1σ uncertainties in Appendices (Table A1) as well as
values of measured equivalent doses for individual aliquots in each sample (Table A2).

OSL ages were analysed with the Bayesian approach to modelling the chronology of the sediments sequence, which uses the
lithostratigraphic record and numerical age of sediments. The *prior* model consists of sequence of sediments units arranged
in stratigraphic order and inferred from lithostratigraphy. Numerical dating controls (OSL age probability distributions)
constrain the possible time of sediments deposition. In Bayesian analysis, they represent the *likehood* that any one sample
has a particular age. Bayesian age modelling was performed using *Sequence* algorithms in OxCal (Bronk Ramsey, 2009a),
ver. 4.4. The algorithms use Markov chain Monte Carlo (MCMC) sampling to build a distribution of possible solutions and
to generate a probability called the *posterior* density estimate for each sample. It is a combination of both the *prior* model
and the *likehood* probability. These density estimates take into account the lithostratigraphic order (*prior*) and typically
reduce the uncertainty range in comparison to *likehood* probabilities. The *Sequence* model in OxCal was divided into a series
of *Phases*, each representing the stages of sediments deposition which may be correlated with particular dating controls.
Thus, each *Phase* consists of a group of dating controls and is separated by *Boundary* commands, which delimit the duration
of each *Phase* and generate an age posterior density estimate. Moreover, we used *Before* ("*terminus ante quem*") command
to constrain the chronology when stages evidently pre-date particular event. The whole *Sequence* is constrained by *Boundary*
commands, which delimit the start and the end of the model. The *Sequence* begins with the *Boundary* "Start" command and





the *Phase* "MIS 6", which consists of three OSL ages of the Rz1 fluvioglacial sediments. Then, the "Rz2a till" was introduced with a *Boundary* command and subsequently the *Phase* "I periglacial phase" consisting of a group of five OSL dating controls from aeolian sand filling fossil periglacial wedges of horizon K1 was defined. The next stage is *Boundary* command "Rz2b till" and above that the *Phase* "II periglacial phase" consisting of three OSL dating controls from aeolian sand filling fossil periglacial wedge of horizon K2 was introduced. The uppermost till layer was defined with the *Boundary* command "Rz2c till" and the constraint was that it had to pre-date (*Before*) radiocarbon age of sample Gd-10818 (16 190 ± 330 cal yr BP – calibrated with *IntCal20* curve in OxCal ver. 4.4) of organic deposits at Stary Cieszyn coring site, because this deposits were very likely formed after deglaciation (Niewiarowski, 2003). The *Sequence* is closed with the *Boundary* "End" command. The notation of commands used to process the algorithms is available in Appendices (Table A3).

### 3.2 $^{10}$Be surface exposure dating

Samples for $^{10}$Be dating were collected from massive and intact boulders resting on glacial landforms. Sampled boulders are large and stable (embedded into the ground) granitic rocks protruding above the ground surface (Fig. 4b). Samples were taken with a manual jackhammer or with a hammer and chisel from the upper surface of boulders. All boulders are characterized by quartz-rich lithologies as granitoids, granite gneisses and gneisses, thus 150–200 g of material per sample was enough for further preparation.

#### 3.2.1 New samples

The first stages of 'LUB' samples (n = 5) preparation were conducted at the laboratory of the University of Gdańsk, Poland. Samples were crushed and sieved; the 0.25-0.71 mm quartz fraction was decontaminated by heavy liquid (SPT) separation (to remove heavy minerals) and froth flotation (to remove feldspars). Then successive acid leaching (2% HF + HNO$_3$) in a hot ultrasonic bath was applied in order to purify quartz. The purity of quartz was checked with ICP-OES analysis for Al content. The next stages of preparation were conducted at the Cosmonuclide Laboratory at the Laboratoire de Géographie Physique (LGP) in Meudon, France. Purified quartz was spiked systematically with ~460 mg of a commercial $^9$Be carrier solution (concentration of 998 mg/l ± 3.7 mg/l) and then dissolved with concentrated HF. Beryllium was separated from remaining metals and purified in three stages: (1) anion column to remove Fe(III), (2) cation column to remove Ti, alkalis and separate Be from Al, and (3) hydroxide precipitation to remove residual alkalis, Mg and Ca. Samples were then dried, oxidized and mixed with niobium powder before being pressed in cathodes for AMS measurements. The $^{10}$Be/$^9$Be ratios were measured by accelerator mass spectrometry (AMS) at the French National AMS Facility ASTER, Aix-en-Provence (Arnold et al., 2010). The measured $^{10}$Be/$^9$Be ratios were normalized relative to the in-house standard STD-11 using an assigned $^{10}$Be/$^9$Be ratio of $(1.191 \pm 0.013) \times 10^{-11}$ (Braucher et al., 2015) and a $^{10}$Be half-life of $(1.387 \pm 0.012) \times 10^{-6}$ years (Chmeleff et al., 2010; Korschinek et al., 2010). Analytical 1σ uncertainties include uncertainties in AMS counting statistics, uncertainty in the standard $^{10}$Be/$^9$Be, an external AMS error of 0.5% (Arnold et al., 2010), and a chemical blank measurement.



[10]Be ages were calculated using the most recent global production rate (Borchers et al., 2016) and the time dependent scaling
scheme for spallation according to Lal (1991) and Stone (2000) (the 'Lm' scaling scheme). We corrected the [10]Be production
rate for sample thickness according to an exponential function (Lal, 1991) and assuming an average density of 2.7 g/cm$^3$ for
granitoid, granite gneiss and gneiss. The appropriate correction for self-shielding (boulder geometry) was applied when the
surface of the sampled boulder had a slope of more than 10°. No correction for surface erosion of boulders was applied, as
we interpret the [10]Be results as minimum ages. All calculations were performed using the online exposure age calculator
formerly known as the CRONUS-Earth online exposure age calculator – version 3 (http://hess.ess.washington.edu/math/;
accessed: 10.05.2023.), which is an updated version of the online calculator described by (Balco et al., 2008). Ages are
reported with 1σ uncertainties (including analytical uncertainties and the production rate uncertainty) in Table A4 in
Appendices.

**3.2.2 Recalculated samples**

We recalculated [10]Be ages already published for the study area (n = 9) based on data available in Rinterknecht et al. (2005,
2006) and Tylmann et al. (2019). We followed the same procedure of exposure age calculations as described in the above
section. Recalculated [10]Be exposure ages are also reported with 1σ uncertainties (including analytical uncertainties and the
production rate uncertainty) in Table A4 (see Appendices).

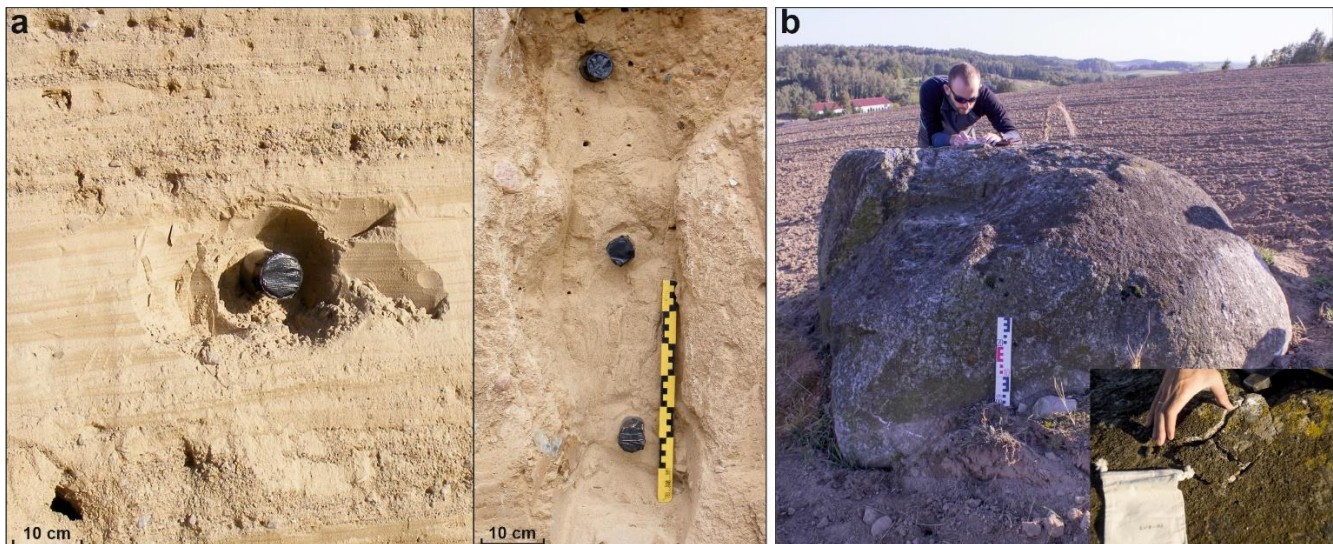

**Figure 4:** (a) Sampling for OSL dating in Rz1 unit (left picture) and K2 periglacial wedge (right picture). Sandy deposits were sampled
with plastic tubes protected with black PCV tape. (b) Sampling for [10]Be dating. Upper surface of erratic boulder was sampled with
a hammer and chisel.



## 4. Results

### 4.1 OSL ages

Three OSL samples from unit Rz1 reveal various distributions of equivalent doses measured for individual aliquots. Sample GdTL-1351 has the most clustered, unimodal distribution with σOD = 12%. Samples GdTL-1352 and GdTL-1353 have bimodal and trimodal distributions respectively, with σOD parameters over 20% (Fig. 5a). For all three samples MAM model was applied to calculate $D_e$ value, as Rz1 unit consists of fluvioglacial poorly sorted sediments which may contain populations of partially bleached grains. It is visible especially within aliquots distribution of samples GdTL-1352 and

GdTL-1353. OSL ages calculated for Rz1 sediments are: 148.9 ± 7.2 ka (GdTL-1351), 143.9 ± 9.1 ka (GdTL-1352) and 126.8 ± 11.0 ka (GdTL-1353). Therefore, depositions of Rz1 unit during the cold Marine Isotope Stage (MIS) 6 is the most likely.

   Deposits filling two fossil periglacial wedges of horizon K1 were sampled for OSL dating. Two samples (Gd-TL-1349 and GdTL-1350) were taken from one wedge, and three samples (GdTL-1879, GdTL-1880 and GdTL-1881) were taken from

another one (Fig. 5a). Distributions of equivalent doses measured for individual aliquots in these samples are well clustered with σOD parameters below 20% (from 5% to 16%). Unimodal distributions dominates and only one sample (GdTL-1350) revealing bimodal type of probability curve. $D_e$ values were determined with CAM model, and OSL ages calculated for aeolian sand filling K1 wedges are: 18.7 ± 1.0 ka (GdTL-1349), 14.5 ± 0.8 ka (GdTL-1350), 19.1 ± 0.9 ka (GdTL-1879), 18.2 ± 0.8 ka (GdTL-1880) and 17.4 ± 0.9 ka (GdTL-1881). Therefore, the most likely timing of sand deposition within K1

periglacial wedges is around 20–17 ka ago, and it may be correlated with MIS 2.

   Three OSL samples were taken from the fossil periglacial sand wedge K2 (GdTL-1346, GdTL-1347 and GdTL-1348). Distributions of equivalent doses measured for individual aliquots in these samples are also well clustered and unimodal with σOD parameter from 9% to 11% (Fig. 5a). $D_e$ values were determined with CAM model, and OSL ages calculated for aeolian sand filling K2 wedge are: 18.5 ± 0.9 ka (GdTL-1346), 16.4 ± 0.8 ka (GdTL-1347) and 17.3 ± 0.8 ka (GdTL-1348).

The most likely timing of sand deposition is around 19–16 ka ago, so it may be correlated with MIS 2.

### 4.2 Bayesian modelling

   The run of the *Sequence* model was conducted in the *Outlier* mode, which assumes that outliers are distributed according to a student T distribution with five degrees of freedom; the scale is allowed to lie anywhere between $10^0$ to $10^4$ years (Bronk Ramsey, 2009b). In the initial model, the dating controls were all entered with a prior probability of 0.05 of being an outlier.

Ages having an agreement index with the initial model <60%, and exceeding the 0.05 threshold of probability of being outliers in the initial model results, were down-weighted by being assigned an adequate higher prior probability of being outliers. Then, a re-run of the same *Sequence* model was conducted for the chronological sequence with down-weighted ages. Finally, the agreement index for the re-run model ($A_{model}$) was used to evaluate the reliability of the chronological



sequences obtained (Chiverrell et al., 2013). Both input ages and modelled ages were reported with 1σ uncertainty (68.2% probability).

**Figure 5:** (a) Sediments profile of the entire exposed sequence with sedimentary features, OSL ages and distributions of equivalent doses measured for individual aliquots. (b) Spatial distribution of sampled boulders with [10]Be ages. New samples are indicated with white dots while recalculated samples are indicated with dark dots. Ages identified as outliers are marked in red, accepted ages are marked in blue.
All ages are given in ka. Inset graph shows distribution of [10]Be ages with kernel density estimate curve and statistics before (red) and after (blue) excluding outliers. Bayesian ages of till layers Rz2a, Rz2b and Rz2c are also marked (blue lines) for comparison with [10]Be ages.

The initial model based on the assumed sequence of events and all dating controls showed a rather poor agreement index (41.3%), which suggested that the results of the initial *Sequence* were not reliable and some outliers and problematic ages must occur among the dating controls. We identified an outlier with the individual agreement index <10%. One OSL age
belonging to the *Phase* "I periglacial phase" (sample GdTL-1350) showed a low agreement index of 4.6% and the



probability of being an outlier was estimated at 85% by the model. The age of this sample is 14.5 ± 0.8 ka, and it is most probably too young for the periglacial horizon K1. Thus, it was down-weighted by being assigned a prior probability of 0.85 of being outliers in the re-run model, which showed a much better agreement index (103.4%). The individual agreement index for the modeled ages ranges between 75.2% and 128.2%, which means that the model is consistent and reliable. The

modeled age distribution for the Rz2a till is 19.2 ± 1.1 ka, for the Rz2b till is 17.8 ± 0.5 ka and for the Rz2c till is 16.9 ± 0.5 ka (Table A5 in Appendices). The results show that timing of the ice advances associated with Rz2a, Rz2b and Rz2c basal tills may be constrained to millennial-scale cycles of the palaeo-ice margin fluctuations at ~19 ka, ~18 ka and ~17 ka.

### 4.3 $^{10}$Be ages

Surface exposure ages of boulders located in the study area range between 5.8 ± 0.8 ka and 40.3 ± 3.9 ka (Fig. 5b).

Distribution of ages (n = 14) is polymodal with the main mode occurring at ~18 ka. The reduced chi-squared test indicates that the ages are poorly clustered: $\chi^2_R = 35.25$. We identify two of the oldest ages (40.3 ± 3.9 ka and 35.5 ± 3.7 ka) and one of the youngest ages (5.8 ± 0.8 ka) as deviating the most from the main mode. They do not fall into a confidence interval arithmetic average ± 1.5 × IQR (interquartile range, which is the range between the third quartile – Q3 and the first quartile – Q1 of the population), and are thus identified as outliers. For the boulders that are "too old", they most probably contain

beryllium inherited from episodes of exposure pre-dating the last deglaciation, and the "too young" age may be a result of boulder exposition after deglaciation and/or significant postglacial erosion of sampled surface. After excluding these outliers, the remaining eleven ages range between 12.5 ± 1.2 ka and 25.8 ± 2.4 ka and reduced chi-squared test shows a much improved cluster: $\chi^2_R = 6.31$. However, the variability of the remaining ages is 24.1%, and with a $\chi^2_R > 2$, the dataset can be described as poorly-clustered (Blomdin et al., 2016). The arithmetic mean and the standard deviation for these eleven surface

exposure ages is 18.0 ± 4.3 ka (Fig. 5b), and it could represent the minimum deglaciation age of the study area, however geomorphological processes could have had large impact on the spread of exposure ages (Heyman et al., 2011).

## 5. Discussion

### 5.1 Timing and dynamics of the last FIS oscillations

The first ice sheet advance which deposited Rz2a till dated at 19.2 ± 1.1 ka corresponds most likely to the local Last Glacial

Maximum (LGM) ice advance associated with the maximum expansion of the last FIS in this region (Fig. 6a). The age of the local LGM in north-central and north-eastern Poland was recently estimated at ~19.0–18.5 ka based on OSL dating and re-interpretation of available cosmogenic ages (Wysota et al., 2009, Marks, 2012) or at the most likely time interval 22–18 ka, based on new cosmogenic chronology interpreted together with available radiocarbon and luminescence ages (Tylmann et al., 2019). After maximum expansion of the last FIS, the ice margin retreated and periglacial conditions with frost

contraction of the exposed ground surface and aeolian deposition of sand occurred, leading to the formation of periglacial horizon K1 at Rożental site (Fig. 6b). Moreover, mass movements and denudation processes were also active at this stage,





which is indicated by a partly eroded Rz2a till layer and the gravitational deformation of fossil sand wedges of horizon K1 (Tylmann et al., 2014).



**Figure 6:** Reconstruction of palaeo-ice margin oscillations in the study area and millennial cycles recorded in marine sediments from the eastern edge of the North Atlantic with ice core record from Greenland. a) Maximum extent of the last FIS around 19 ka ago. (b) Ice-free conditions around 19-18 ka ago. (c) Ice sheet advance around 18 ka ago. (d) Ice-free conditions around 18-17 ka ago. (e) Ice sheet advance around 17 ka ago. (f) Variations of $\delta^{18}O$ signature in NGRIP ice core (NGRIP-members, 2004), $\varepsilon_{Nd}$ isotopic composition, turbidite frequency and sediments load in marine sediments from the Bay of Biscay (for details see Toucanne et al., 2008, 2010 and 2015). Episodes
of the Channel River large meltwater discharge (R-events) are marked as well as Heinrich event 1 (HE 1) and the most likely age of tills Rz2a, Rz2b and Rz2c.

The second ice advance deposited Rz2b till dated at 17.8 ± 0.5 ka. The extent of this ice advance is not unequivocally determined, however the ice most likely covered the locality of the Rożental site. Based on spatial distribution of glacial landforms and sediments, i.e. outlets of subglacial valleys and proximal edges of narrow outwash plains located on south and
south-eastern slopes of the Lubawa Upland (Fig. 2), we argue that this ice advance could cover the highly elevated central part of the study area. The ice margin probably reached south-eastern and eastern slopes of the Lubawa Upland (Fig. 6c). After ~18 ka the ice sheet retreated again and the minimum deglaciation age of the study area inferred from surface exposure



dating of boulders (18.0 ± 4.3 ka) probably represents this stage of the ice margin oscillations. However, the scatter of [10]Be ages is large (from 12.5 ± 1.2 ka to 25.8 ± 2.4 ka after excluding outliers) and various factors, such as: inherited [10]Be signal,

redeposition of boulders, degradation of moraines and erosion of boulders surface, have had significant impact on the spread of reported exposure ages (Heyman et al., 2011; Blomdin et al., 2016). [10]Be age of boulder located in the vicinity of the study area and most likely in the same morphostratigraphic zone (sample LGM-11, 17.5 ± 1.6 ka), also suggests ice margin recession in this region immediately after ~18 ka (Fig. 1b). The ice margin retreated to the north and north-west of the study area and periglacial conditions occurred again, at least in the locality of the Rożental site where periglacial wedge K2 was

formed (Fig. 6d).

The third ice advance which deposited the R2c till at Rożental (16.9 ± 0.5 ka), was probably the least extensive (Fig. 6e). The ice sheet covered only north-western edge of the Lubawa Upland and the ice margin was probably located along the ice-marginal valleys (Fig. 2a), which drained glacial meltwater south-westwards. The final deglaciation of the study area occurred after ~17 ka.

**5.2 Correlation with regional glacial phases**

Our results indicate millennial-scale oscillations of the last FIS in northern Poland between ~19 and ~17 ka. These cycles of ice sheet advances and retreats occurred in the late stage of the local LGM and during the subsequent deglaciation of this region. Timing of the maximum extent of the last FIS in its southern sector was recently constrained to ~24–23 ka in western Poland and north-eastern Germany during the Brandenburg (Leszno) Phase and to ~19 ka in north-central and north-eastern

Poland during the Frankfurt (Poznań) Phase (Wysota et al., 2009; Ehlers et al., 2011; Marks, 2012; Marks et al., 2022). Based on the recent [10]Be dating and comparison to the available radiocarbon and luminescence chronology, the most likely time intervals for the local LGM are 25–21 ka in western Poland and 22–18 ka in north-eastern Poland (Tylmann et al., 2019). Therefore, the Rz2a till dated at 19.2 ± 1.1 ka is correlated with the local LGM ice advance. In north-central Poland timing of the ice advance which reached the maximum limit of the last FIS was constrained based on OSL dating of the

Upper Weichselian glacial sequence to ~18.5 ka (Wysota et al., 2009), which might also be correlated with our age constraint for Rz2a till (19.2 ± 1.1 ka). However, based on apparent OSL ages obtained for Rz1 unit and periglacial wedges of horizon K1, the possible time window for deposition of Rz2a till is wide and ranges between ~150 ka and ~19–17 ka (Fig. 5a). This suggests that Rz2a basal till may be correlated with MIS 6 ice advance (the Late Saalian glaciation) or with MIS 4/MIS 2 ice advances occurring before 19–17 ka. We argue that sediments of Rz1 unit the most likely represents recession

phase of the MIS 6 ice sheet, as they consists of relatively coarse-grained, horizontally bedded lithofacies associated with intensive ablation cycles. Therefore, in our opinion deposition of the Rz2a till after MIS 6 is the most likely and our modelling, which takes into account the whole sedimentary sequence, shows the most probable age of this till correlated with MIS 2 (19.2 ± 1.1 ka).



The second ice advance constrained to 17.8 ± 0.5 ka is comparable to one of the ice-marginal formation formed during the
last deglaciation in north-eastern corner of Poland (Łopuchowo 2 and Gulbieniszki moraines) and dated at 17.9 ± 1.3 ka with
cosmogenic $^{36}$Cl (ages reported in Dzierżek and Zreda, 2007). This ice advance may be also related to regional sub-phase
distinguished in northern Poland based on geomorphology between maximum extent of the last FIS and the Pomeranian
Phase – the Kujawy-Dobrzyń subphase (Kozarski, 1995; Niewiarowski et al., 1995). However, a broad correlation along the
palaeo-ice margin over a large distances is impeded and uncertain, as various sections of ice-marginal formations at the
southern fringe of the last FIS were usually formed asynchronously (e.g., Dzierżek and Zreda, 2007; Lüthgens and Böse,
2012; Tylmann et al., 2022).

The last ice advance in the study area was dated at 16.9 ± 0.5 ka and may be correlated with the Pomeranian Phase of the last
deglaciation, which age was recently constrained to 17–16 ka (e.g., Marks, 2012, Marks et al., 2022) or 18–17 ka (Stroeven
et al., 2016). However, new studies showed that the age of ice-marginal formations at the southern fringe of the last FIS
traditionally correlated with a discrete time interval during the Pomeranian Phase, covers in-fact a wide time window
between 20 ka and 15 ka (Tylmann et al., 2022). Thus, we argue that ~17 ka and ice re-advance occurred on the north-
western slope of the Lubawa Upland, and this re-advance could be correlated with ice advances and/or ice margin stillstands
within the Mazury Ice Stream which are dated at 18–17 ka (Tylmann et al. 2022).

**5.3 Millennial-scale fluctuations of the last FIS**

Our results show very dynamic oscillations of one particular segment of the FIS's southern front. The last FIS advanced and
retreated over a relatively short period of time (2–3 thousands of years), leaving lithostratigraphic record (basal tills) of three
ice advances at a millennial-scale cycle: ~19 ka, ~18 ka and ~17 ka. Millennial-scale fluctuations of the southern fringe of
the last FIS have been already explored based on linking properties of marine deposits from the eastern edge of North
Atlantic precisely constrained by a radiocarbon chronology, with dynamics of the terrestrial palaeo-ice sheet margin in
Europe (e.g., Zaragossi et al, 2001, 2006; Toucanne et al., 2008, 2010, 2015). During the last deglaciation, meltwater from
the southern front of the FIS transported terrigenous deposits along the Channel River network (including ice-marginal
valleys system – urstromtal – in the North European Plain) towards the Bay of Biscay. It was a key depocenter for far-
travelled sediments released from the European ice sheets, including the southern FIS. Properties of sediments sequences
deposited in the Bay of Biscay, such as turbidite deposits frequency (Zaragossi et al, 2006; Toucanne et al., 2008) or
sediments accumulation ratio (Toucanne et al., 2010), indicate increased meltwater discharge and enhanced ice sheet decay
between ~20 and ~17 ka. After 20 ka sediments loading within the Bay of Biscay depocenter rose significantly in
comparison to lower sediments accumulation ratio and turbidity activity between ~30 ka and ~20 ka (Fig. 6f). Between ~19
ka and 18.5 ka there is a sudden reduction of turbidity activity (Fig. 6f), however this could be a result of the first well-
known abrupt sea level rise – meltwater pulse at ~19 ka – 19-ka MWP (Clark et al., 2004). This reflects a significant retreat
of the southern FIS ice margin after the LGM period, which in our results is indicated after the first ice advance dated at 19.2





± 1.1 ka. Maximum turbidity activity and sediment load, which occurred at ~18.3–17.0 ka, corresponds to the main phase of the FIS melting in the North European Plain (Toucanne et al., 2008). The latter could be roughly correlated with an ice margin retreat after the second ice advance in our study area, dated at 17.8 ± 0.5 ka. After ~17.5–17.0 ka the meltwater discharge from the southern FIS significantly decreased in response to the initiation of a deglacial pause and a global re-advance of glaciers and ice sheets in Europe corresponding to Heinrich event 1 (HE1) (Zaragossi et al., 2001; Toucanne et al., 2009). This event is correlated to the last ice advance recorded in our study area and dated at 16.9 ± 0.5 ka (Fig. 6f).

Coupling between the southern FIS fluctuations and the Channel River meltwater discharge was also investigated by Toucanne et al. (2015), who used neodymium isotopic composition of sediments, a powerful tracer for terrigenous sediments geographical provenance, cored from the Bay of Biscay seafloor and sampled from moraines, ice-marginal valleys and proglacial lakes alongside the FIS southern margin. As a results, episodes of the Channel River large meltwater discharges (R-events) were distinguished and correlated with the FIS dynamics (Fig. 6f). The first ice advance in our study area (19.2 ± 1.1 ka) corresponds to the millennial-scale intervals of Channel River shutdowns (i.e. pauses in deglaciation) between 21.3 ± 0.2 ka (i.e., end of the R3 event) and 20.3 ± 0.2 ka (i.e., onset of the R4 event) or between 18.7 ± 0.3 ka (i.e., end of the R4 event) and 18.2 ± 0.2 ka (i.e., onset of the R5 event). The second ice advance in our study area, constrained to 17.8 ± 0.5 ka, falls within an episode of substantial ice marginal retreat recorded by Toucanne et al. (2015) between 18.2 ± 0.2 ka and 16.7 ± 0.2 ka (R5 event, just before HE1). However, the third ice-advance in our study area (16.9 ± 0.5 ka) correlates well with a pause in the overall ice margin retreat between 16.7 ± 0.2 ka and 15.7 ± 0.3 ka (HE1) according to Toucanne et al. (2015) or between 17.2 ± 0.4 ka and 15.7 ± 0.3 ka according to reconstruction of the FIS dynamics in the East European Plain (Soulet et al. 2013).

## 6. Conclusions

Our results indicate millennial-scale oscillations of the last FIS in northern Poland between ~19 and ~17 ka. Based on combined luminescence and [10]Be dating we show that the last FIS advanced and retreated over a relatively short period of time (2–3 thousands of years), leaving lithostratigraphic record (basal tills) of three ice re-advances at a millennial-scale cycle: 19.2 ± 1.1 ka, 17.8 ± 0.5 ka and 16.9 ± 0.5 ka. This is the first terrestrial record of millennial-scale palaeo-ice margin oscillations at the southern fringe of the FIS during the last glacial cycle. Cycles of ice sheet re-advances and retreats occurred in the late stage of the local LGM and during the subsequent deglaciation of this region. The first ice re-advance which deposited the Rz2a till (19.2 ± 1.1 ka) is correlated with the local LGM ice advance. The second ice re-advance constrained to 17.8 ± 0.5 ka (Rz2b till) is comparable to one of the ice-marginal formation deposited in north-eastern region of Poland and dated at 17.9 ± 1.3 ka with cosmogenic [36]Cl. The last ice re-advance dated at 16.9 ± 0.5 ka (Rz2c till) is correlated with ice advances and/or ice margin stillstands in the Mazury Ice Stream during the Pomeranian Phase.

Millennial-scale palaeo-ice margin oscillations at the southern fringe of the FIS inferred from terrestrial record was linked to cycle recorded in marine deposits with precisely constrained radiocarbon chronology from the eastern edge of the North
Atlantic. The first ice advance was dated at 19.2 ± 1.1 ka and is reflected in a sudden reduction of turbidity activity between ~19 ka and 18.5 ka recorded in marine sediments from the Bay of Biscay. The subsequent ice margin retreat is reflected in the first well-known abrupt sea level rise – meltwater pulse at ~19 ka (19-ka MWP). A second ice re-advance in our study area was dated at 17.8 ± 0.5 ka. The following ice margin retreat is roughly correlated with the maximum turbidity activity and sediment load at ~18.3–17.0 ka in the Bay of Biscay. The third and last ice re-advance recorded in our study area was dated at 16.9 ± 0.5 ka and corresponds to a significant drop of meltwater discharge from the southern FIS, the initiation of a deglacial pause and a global re-advance of glaciers and ice sheets in Europe related to the HE1.

**Data Availability**

The data that support findings of this study are available upon the reasonable request.

**Team list**

Georges Aumaître, Didier L. Bourlès and Karim Keddadouche (Aix Marseille Univ, CNRS, IRD, INRAE, CEREGE, Aix-en-Provence, 13545, France).

**Author Contribution**

KT was responsible for conceptualisation of the study, fieldwork, sample preparation for $^{10}$Be dating, data analysis and interpretation, figures preparation, writing and editing of the manuscript. WW contributed to the fieldwork, sampling for OSL dating, analysing and interpreting data, editing and proof-reading of the manuscript. VR was responsible for sample preparation for $^{10}$Be dating, contributed in editing and proof-reading of the manuscript. PM was responsible for OSL dating and contributed to data analysis and proof-reading of the manuscript. ABG was also responsible for ICP-OES analysis. ASTER Team preformed AMS measurements of $^{10}$Be/$^{9}$Be ratios.

**Competing interests**

The corresponding author has declared that none of the authors has any competing interests.

**Acknowledgements**

This work was supported by the National Science Centre in Poland [grant no. 2011/01/N/ST10/05880 to KT], the Polish Ministry of Science and Higher Education [grant no. N N306 316835 to WW] and funding from the Faculty of Oceanography and Geography, University of Gdańsk. The ASTER AMS national facility (CEREGE, Aix-en-Provence) is supported by the INSU/CNRS, the ANR through the "Projets thematiques d'excellence" program for the "Equipements d'excellence" ASTER-CEREGE action and IRD.



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



## Appendices

**Table A1:** OSL samples with laboratory data and parameters used during OSL age calculation.

| Lab. Code | Th (Bq/kg) | U (Bq/kg) | K (Bq/kg) | $D_r$ (Gy/ka) | σOD (%) | $D_e$ (Gy) | OSL age (ka) |
|---|---|---|---|---|---|---|---|
| GdTL-1346 | 8.7 ± 0.4 | 10.3 ± 0.3 | 295 ± 13 | 1.25 ± 0.05 | 9 | 23.1 ± 0.7 | 18.5 ± 0.9 |
| GdTL-1347 | 9.7 ± 0.6 | 12.9 ± 0.4 | 320 ± 15 | 1.38 ± 0.06 | 11 | 22.7 ± 0.7 | 16.4 ± 0.8 |
| GdTL-1348 | 9.8 ± 0.5 | 12.0 ± 0.4 | 314 ± 15 | 1.36 ± 0.06 | 10 | 23.8 ± 0.7 | 17.3 ± 0.8 |
| GdTL-1349 | 8.2 ± 0.6 | 10.4 ± 0.4 | 297 ± 16 | 1.27 ± 0.06 | 11 | 23.2 ± 0.7 | 18.7 ± 1.0 |
| GdTL-1350 | 10.9 ± 0.5 | 13.3 ± 0.4 | 340 ± 15 | 1.44 ± 0.06 | 16 | 21.1 ± 0.8 | 14.5 ± 0.8 |
| GdTL-1351 | 8.3 ± 0.4 | 8.9 ± 0.4 | 330 ± 10 | 1.26 ± 0.05 | 12 | 186.5 ± 5.5 | 148.9 ± 7.2 |
| GdTL-1352 | 14.1 ± 0.5 | 12.0 ± 0.4 | 344 ± 10 | 1.45 ± 0.05 | 25 | 209.0 ± 11.3 | 143.9 ± 9.1 |
| GdTL-1353 | 9.7 ± 0.5 | 9.4 ± 0.4 | 318 ± 10 | 1.30 ± 0.05 | 24 | 165.3 ± 12.8 | 126.8 ± 11.0 |
| GdTL-1879 | 7.3 ± 0.3 | 5.9 ± 0.2 | 327 ± 10 | 1.25 ± 0.05 | 10 | 24.0 ± 0.8 | 19.1 ± 0.9 |
| GdTL-1880 | 7.5 ± 0.3 | 6.4 ± 0.2 | 327 ± 10 | 1.27 ± 0.05 | 5 | 23.2 ± 0.5 | 18.2 ± 0.8 |
| GdTL-1881 | 7.5 ± 0.3 | 6.8 ± 0.3 | 332 ± 10 | 1.35 ± 0.05 | 12 | 23.6 ± 0.8 | 17.4 ± 0.9 |

**Table A2:** Equivalent doses ($D_e$) measured for individual aliquots in OSL samples. All values are given in Gy.

| GdTL-1346 (n = 20) | | GdTL-1347 (n = 20) | | GdTL-1348 (n = 20) | | GdTL-1349 (n = 22) | | GdTL-1350 (n = 24) | | GdTL-1351 (n = 31) | | GdTL-1352 (n = 19) | | GdTL-1353 (n = 28) | | GdTL-1879 (n = 23) | | GdTL-1880 (n = 22) | | GdTL-1881 (n = 23) | |
|---|---|---|---|---|---|---|---|---|---|---|---|---|---|---|---|---|---|---|---|---|---|
| $D_e$ | ± | $D_e$ | ± | $D_e$ | ± | $D_e$ | ± | $D_e$ | ± | $D_e$ | ± | $D_e$ | ± | $D_e$ | ± | $D_e$ | ± | $D_e$ | ± | $D_e$ | ± |
| 19.07 | 2.00 | 19.83 | 1.50 | 20.79 | 1.97 | 16.61 | 1.85 | 26.53 | 2.00 | 134.76 | 13.48 | 172.30 | 17.23 | 148.38 | 14.84 | 18.50 | 1.50 | 20.87 | 1.50 | 19.58 | 1.50 |
| 19.15 | 2.00 | 22.25 | 1.50 | 21.34 | 1.50 | 18.33 | 1.50 | 24.04 | 2.00 | 140.90 | 14.09 | 183.50 | 18.35 | 148.40 | 14.84 | 19.72 | 1.50 | 20.95 | 1.50 | 20.03 | 1.50 |
| 19.31 | 2.00 | 23.84 | 1.50 | 21.54 | 1.50 | 18.55 | 1.50 | 26.14 | 2.00 | 143.30 | 14.33 | 183.66 | 18.37 | 148.41 | 14.84 | 20.99 | 1.50 | 22.12 | 1.50 | 20.22 | 1.50 |
| 19.40 | 2.00 | 23.96 | 1.50 | 23.50 | 1.50 | 19.61 | 1.50 | 22.07 | 2.00 | 152.70 | 15.27 | 184.60 | 18.46 | 152.55 | 15.26 | 21.05 | 1.50 | 22.25 | 1.50 | 21.59 | 1.50 |
| 22.06 | 2.00 | 24.18 | 1.50 | 23.52 | 1.50 | 19.79 | 1.50 | 24.83 | 2.00 | 157.00 | 15.70 | 196.35 | 19.64 | 154.80 | 15.48 | 21.91 | 1.80 | 22.33 | 1.50 | 21.83 | 1.50 |
| 22.72 | 2.00 | 24.78 | 1.50 | 23.66 | 1.50 | 21.88 | 1.50 | 21.11 | 2.00 | 167.05 | 16.71 | 203.57 | 20.36 | 160.80 | 16.08 | 21.91 | 1.50 | 22.53 | 1.50 | 22.95 | 1.50 |
| 22.72 | 2.00 | 24.81 | 1.50 | 23.82 | 1.50 | 25.09 | 1.50 | 18.62 | 2.00 | 167.08 | 16.71 | 210.27 | 21.03 | 160.80 | 16.08 | 23.30 | 1.50 | 22.69 | 1.50 | 23.09 | 1.50 |
| 23.04 | 2.00 | 25.01 | 1.50 | 23.97 | 1.50 | 25.10 | 1.50 | 21.37 | 2.00 | 170.60 | 17.06 | 211.91 | 21.19 | 161.04 | 16.10 | 23.31 | 1.50 | 23.10 | 1.50 | 23.43 | 1.50 |
| 24.93 | 2.00 | 26.76 | 1.50 | 27.08 | 1.50 | 26.87 | 1.50 | 18.39 | 2.00 | 172.18 | 17.22 | 212.96 | 21.30 | 162.14 | 16.21 | 24.38 | 1.50 | 23.45 | 1.50 | 23.43 | 1.50 |
| 27.49 | 2.00 | 23.13 | 1.50 | 27.97 | 1.50 | 28.01 | 2.00 | 15.29 | 1.50 | 173.97 | 17.40 | 215.50 | 21.55 | 168.14 | 16.81 | 24.43 | 1.80 | 24.24 | 1.50 | 23.64 | 1.50 |
| 28.12 | 2.00 | 19.51 | 1.50 | 27.25 | 1.50 | 30.39 | 4.83 | 15.48 | 1.50 | 174.40 | 17.44 | 225.30 | 22.53 | 182.04 | 18.20 | 24.70 | 1.80 | 24.64 | 1.50 | 24.08 | 1.50 |
| 21.25 | 2.00 | 24.57 | 1.50 | 27.46 | 1.50 | 21.09 | 1.50 | 15.61 | 1.50 | 177.40 | 17.74 | 227.35 | 22.74 | 188.13 | 18.81 | 24.79 | 1.50 | 24.86 | 1.50 | 24.17 | 1.50 |
| 24.46 | 2.00 | 17.43 | 1.50 | 19.71 | 1.50 | 23.65 | 1.50 | 17.37 | 1.50 | 178.07 | 17.81 | 230.61 | 23.06 | 201.60 | 20.16 | 25.18 | 1.80 | 24.87 | 1.50 | 24.20 | 1.50 |
| 24.99 | 2.00 | 25.55 | 1.50 | 24.38 | 1.50 | 23.40 | 1.50 | 17.64 | 1.50 | 183.60 | 18.36 | 249.00 | 24.90 | 201.60 | 20.16 | 25.28 | 1.50 | 24.89 | 1.50 | 24.24 | 1.50 |
| 27.44 | 2.00 | 18.30 | 1.50 | 19.55 | 1.50 | 25.45 | 2.00 | 18.37 | 1.50 | 186.10 | 18.61 | 262.18 | 26.22 | 205.29 | 20.53 | 25.30 | 1.80 | 25.04 | 1.50 | 25.00 | 1.50 |
| 25.56 | 2.00 | 26.71 | 1.50 | 19.89 | 1.50 | 23.24 | 1.50 | 18.78 | 1.50 | 187.65 | 18.77 | 281.90 | 28.19 | 208.20 | 20.82 | 25.97 | 1.80 | 25.30 | 1.50 | 25.41 | 1.50 |
| 21.81 | 2.00 | 21.87 | 1.50 | 22.54 | 1.50 | 27.30 | 2.00 | 19.17 | 1.50 | 188.93 | 18.89 | 379.80 | 37.98 | 209.70 | 20.97 | 26.21 | 1.80 | 25.37 | 1.50 | 26.23 | 1.50 |
| 24.27 | 2.00 | 20.91 | 1.50 | 22.49 | 1.50 | 22.20 | 1.50 | 19.26 | 1.50 | 189.47 | 18.95 | 414.40 | 41.44 | 209.81 | 20.98 | 26.58 | 1.80 | 25.54 | 1.50 | 26.76 | 1.50 |
| 20.13 | 2.00 | 22.33 | 1.50 | 26.42 | 1.50 | 26.53 | 2.00 | 21.46 | 1.50 | 190.75 | 19.08 | 443.90 | 44.39 | 233.27 | 23.33 | 26.69 | 1.80 | 26.07 | 1.50 | 27.27 | 1.50 |
| 21.27 | 2.00 | 17.76 | 1.50 | 26.77 | 1.50 | 24.04 | 2.00 | 22.43 | 1.50 | 192.88 | 19.29 | | | 237.53 | 23.75 | 28.33 | 1.80 | 26.09 | 1.50 | 27.99 | 1.50 |
| | | | | | | 26.14 | 2.00 | 26.26 | 2.00 | 195.83 | 19.58 | | | 239.03 | 23.90 | 28.48 | 1.80 | 26.80 | 1.50 | 28.62 | 1.50 |
| | | | | | | 22.07 | 1.50 | 26.49 | 2.00 | 198.91 | 19.89 | | | 245.30 | 24.53 | 28.73 | 1.80 | 28.67 | 1.50 | 29.37 | 1.50 |
| | | | | | | | | 26.88 | 2.00 | 207.66 | 20.77 | | | 284.18 | 28.42 | 29.59 | 1.80 | | | 30.06 | 1.50 |





| | | | | | | | 26.95 | 2.00 | 212.71 | 21.27 | | | 286.80 | 28.68 | | | | | |
| | | | | | | | | | 214.46 | 21.45 | | | 307.95 | 30.80 | | | | | |
| | | | | | | | | | 220.60 | 22.06 | | | 309.45 | 30.95 | | | | | |
| | | | | | | | | | 220.90 | 22.09 | | | 311.11 | 31.11 | | | | | |
| | | | | | | | | | 222.30 | 22.23 | | | 322.73 | 32.27 | | | | | |
| | | | | | | | | | 231.12 | 23.11 | | | | | | | | | |
| | | | | | | | | | 238.94 | 23.89 | | | | | | | | | |
| | | | | | | | | | 257.70 | 25.77 | | | | | | | | | |

**Table A3:** The notation of commands used to process the *Sequence* algorithms in OxCal.

```
Options()
{
 BCAD = FALSE;
 kIterations = 100;
 PlusMinus = FALSE;
 SD1 = FALSE;
 SD2 = TRUE;
 SD3 = FALSE;
};
Plot()
{
 Outlier_Model("FIS",T(5),U(0.4),"t");
 Sequence("FIS_oscillations")
 {
  Boundary("START");
  Phase("MIS 6")
  {
   C_Date("GdTL-1351", 148900, 7200)
   {
    Outlier(0.05);
   };
   C_Date("GdTL-1352", 143900, 9100)
   {
    Outlier(0.05);
   };
   C_Date("GdTL-1353", 126800, 11000)
   {
    Outlier(0.05);
   };
  };
  Boundary("Rz2a_till");
  Phase("I_periglacial phase")
  {
   C_Date("GdTL-1349", 18700, 1000)
   {
    Outlier(0.05);
   };
   C_Date("GdTL-1350", 14500, 800)
   {
    Outlier(0.05);
   };
   C_Date("GdTL-1879", 19100, 900)
   {
    Outlier(0.05);
   };
   C_Date("GdTL-1880", 18200, 800)
   {
    Outlier(0.05);
   };
   C_Date("GdTL-1881", 17400, 900)
   {
    Outlier(0.05);
   };
```

```
};
Boundary("Rz2b_till");
Phase("II_periglacial_phase")
{
 C_Date("GdTL-1346", 18500, 900)
 {
  Outlier(0.05);
 };
 C_Date("GdTL-1347", 16400, 800)
 {
  Outlier(0.05);
 };
 C_Date("GdTL-1348", 17300, 800)
 {
  Outlier(0.05);
 };
};
Boundary("Rz2c_till");
Before(Age("Gd-10818", 16190, 330));
Boundary("END");
};
};
```


**Table A4:** Surface exposure [10]Be ages of erratic boulders. The list consists of five new [10]Be ages (LUB samples) and nine ages recalculated from the original data of Rinterknecht et al. (2006) and Tylmann et al. (2019). All [10]Be exposure ages are calculated with 'Lm' time-dependent scaling scheme for spallation according to Lal (1991) and Stone (2000) and the global production rate according to Borchers et al. (2016).

| Sample ID | Latitude N DD | Longitude E DD | Elevation (m a.s.l.) | Boulder lithology | Landform | Sample thickness (cm) | Shielding factor[1] | Quartz (g) | [10Be] ($10^4$ at g$^{-1}$) | Age (ka) |
|---|---|---|---|---|---|---|---|---|---|---|
| **New samples** | | | | | | | | | | |
| LUB-01 | 53.5346 | 19.8096 | 192 | granite | moraine | 1.1 | 0.9999 | 11.296 | 13.11 ± 0.74 | 25.8 ± 2.4 |
| LUB-02 | 53.5196 | 19.8593 | 254 | gneiss | moraine | 2.0 | 1.0000 | 20.414 | 9.87 ± 0.46 | 18.4 ± 1.6 |
| LUB-03 | 53.5252 | 19.8643 | 263 | granitic gneiss | moraine | 4.0 | 0.9874 | 14.790 | 7.42 ± 0.47 | 14.1 ± 1.4 |
| LUB-04 | 53.4983 | 19.8601 | 204 | granite | proglacial valley | 3.3 | 1.0000 | 19.685 | 6.31 ± 0.37 | 12.5 ± 1.2 |
| LUB-05 | 53.5383 | 19.9283 | 286 | gneiss | moraine | 1.5 | 0.9976 | 18.573 | 11.58 ± 0.85 | 20.9 ± 2.2 |
| **Recalculated samples[2]** | | | | | | | | | | |
| LES-5 | 53.5792 | 20.0944 | 180 | granite | moraine | 2.0 | 1.0000 | 40.000 | 19.24 ± 1.16 | 40.3 ± 3.9 |
| LES-6 | 53.6006 | 20.0611 | 151 | gneiss | edge of subglacial valley | 2.0 | 1.0000 | 40.000 | 8.08 ± 0.58 | 17.4 ± 1.8 |
| LES-7 | 53.6250 | 20.0417 | 132 | granite | subglacial valley | 2.0 | 1.0000 | 60.509 | 2.64 ± 0.33 | 5.8 ± 0.8 |
| LES-8 | 53.5111 | 19.9000 | 255 | granite | moraine | 2.0 | 1.0000 | 40.001 | 10.14 ± 1.10 | 19.7 ± 2.6 |
| LES-10 | 53.5764 | 19.9417 | 270 | granite | moraine | 2.0 | 1.0000 | 40.007 | 6.78 ± 0.57 | 13.0 ± 1.5 |
| LES-11 | 53.5222 | 19.8375 | 218 | gneiss | moraine | 2.0 | 1.0000 | 39.993 | 7.94 ± 0.77 | 16.0 ± 2.0 |
| LES-12 | 53.5625 | 19.9528 | 275 | granite | moraine | 2.0 | 1.0000 | 40.007 | 8.46 ± 0.70 | 16.1 ± 1.8 |
| LES-13 | 53.5530 | 19.9250 | 302 | granite | moraine | 2.0 | 1.0000 | 40.005 | 19.15 ± 1.33 | 35.5 ± 3.7 |
| LGM-12 | 53.3874 | 19.752 | 130 | granite | moraine | 1.2 | 1.0000 | 15.070 | 11.50 ± 0.53 | 24.1 ± 2.1 |



AMS $^{10}$Be/$^9$Be results are standardized to NIST SRM 4325 (samples LES) and STD-11 (samples LUB). $^{10}$Be/$^9$Be ratios were corrected for a process blank values of $3.80 \times 10^{-15}$ (samples LES). $3.38 \times 10^{-15}$ (sample LGM-12) and $4.44 \times 10^{-15}$ (samples LUB).

[1] Corresponding to self-shielding (direction and angle of surface dip).

[2] Based on original data from Rinterknecht et al (2005. 2006) and Tylmann et al. (2019).

**Table A5:** OSL dating controls and results of the Bayesian age modelling.

| Phase/*Boundary* | Sample | Age (ka) | Initial model ($A_{model}$ = 41.3%) | | Model with down-weighted age ($A_{model}$ = 103.4%) | |
| --- | --- | --- | --- | --- | --- | --- |
| | | | Modeled age (ka) | A index (%) | Modeled age (ka) | A index (%) |
| *Rz2c till* | | | | | **16.9 ± 0.5** | - |
| II periglacial phase | GdTL-1348 | 17.3 ± 0.8 | 17.3 ± 0.5 | 126.6 | 17.3 ± 0.4 | 128.2 |
| | GdTL-1347 | 16.4 ± 0.8 | 17.2 ± 0.4 | 87.9 | 17.3 ± 0.4 | 83.5 |
| | GdTL-1346 | 18.5 ± 0.9 | 17.4 ± 0.5 | 69.4 | 17.5 ± 0.5 | 75.2 |
| *Rz2b till* | | | | | **17.8 ± 0.5** | - |
| I periglacial phase | GdTL-1881 | 17.4 ± 0.9 | 18.1 ± 0.5 | 97.6 | 18.2 ± 0.5 | 94.2 |
| | GdTL-1880 | 18.2 ± 0.8 | 18.3 ± 0.5 | 121.5 | 18.3 ± 0.5 | 123.2 |
| | GdTL-1879 | 19.1 ± 0.9 | 18.5 ± 0.6 | 99.7 | 18.5 ± 0.6 | 102.9 |
| | GdTL-1350* | 14.5 ± 0.8 | 18.2 ± 0.8 | 4.6 | 18.4 ± 0.7 | 108.1 |
| | GdTL-1349 | 18.7 ± 1.0 | 18.4 ± 0.6 | 119.6 | 18.4 ± 0.6 | 122.7 |
| *Rz2a till* | | | | | **19.2 ± 1.1** | - |
| MIS 6 | GdTL-1353 | 126.8 ±11.0 | 126.7 ±10.9 | 100.4 | 126.7 ± 11.0 | 100.4 |
| | GdTL-1352 | 143.9 ± 9.1 | 142.9 ± 8.8 | 102.0 | 142.9 ± 8.8 | 102.1 |
| | GdTL-1351 | 148.9 ± 7.2 | 147.6 ± 7.1 | 100.5 | 147.6 ± 7.1 | 100.5 |

* age identified in the initial *Sequence* as outlier; the age was down-weighted in the re-run *Sequence* to a prior probability of 0.85 of being an outlier