# Peer review of "Millennial-scale fluctuations of palaeo-ice margin at the southern fringe of the last Fennoscandian Ice Sheet"

_The Cryosphere, 2023_

## Author Comment (AC1)

We differentiated with colors our responses for the reviewers comments and questions. While the reviewers comments and questions are indicated in black, our responses are written in green. Response for particular comment immediately below the comment.

**RC1**: 'Comment on tc-2023-117', Anonymous Referee #1, 31 Oct 2023

This manuscript provides a new vantage on Late Pleistocene variability of the southern margin of the Fennoscandian ice sheet, drawing primarily from a suite of OSL ages on periglacial sediments exposed in the Rożental gravel pit. A number of less-compelling beryllium-10 surface-exposure ages augment the chronology, though owing to the considerable scatter in the latter, they provide only background support for the overall glaciologic story. The authors link apparent stepwise retreat of the ice sheet margin with variability in meltwater discharge into the Atlantic via the Fleuve Manche, a robust marine-geologic indicator of terrestrial meltdown. Altogether, this is an intriguing dataset that is presented in a clear and transparent manner, including excellent figures and thorough descriptions of all methodologies. As such, it will make a valuable contribution to the literature and is suitable for publication in *The Cryosphere*. Before I can recommend publication, however, there are a number of issues that need to be tackled; these are not altogether onerous but will require the authors to reconsider some of their interpretations of the data. I list them below (along with general comments) in the order at which they present themselves in the text.

Thank you very much for your careful review and this generally positive opinion about the manuscript! We tried to answer all questions/remarks and correct the manuscript as far as we can.

Abstract: Look out for typographical errors. This is a fine introduction/synthesis of the manuscript but the incomplete English distracts the reader's attention somewhat. The same goes for the rest of the manuscript, which is well written but would benefit from fine tuning by a native English speaker.

We will try to correct the typographical errors and the general language quality of the manuscript. Referee #2 pointed out a several language suggestions and remarks to the particular statements/words in the text. We will correct them all. Moreover, *The Cryosphere* offers English language copy-editing service for final revised papers within the article processing charge, so the manuscript (if accepted) will be carefully checked before the final publication.

Figure 1: Legend is a little confusing. The black circles with values are recalculated ages, yes? What are the black circles without values? It would also strengthen the figure to justify the positions of the LGM and Pomeranian ice margins – whose work confirms that these positions are correct? What about chronology? Without those details it looks a little like guesswork.

Yes, back dots with values are recalculated ages from Rinterknecht et al. 2006 (LES and POM samples) and from Tylmann et al. 2019 (LGM sample) located outside the study area indicated with dashed grey rectangle. Black dots without values here are also recalculated ages, but located within the study area, and their values are presented in Fig. 5b, together with new ages as a part of the results. We will clarify this in the legend and in the figure caption.

We will also provide info with references according to which source the LGM limit and Pomeranian marginal belt were marked (it was done based on detailed geologic maps), and indicate the possible timing for the LGM limit and Pomeranian recessional phase in this region according to literature and previous research.

Line 110: Such things are subjective. How big is massive? Give approx. size.

OK, we will specify that we are talking about boulders with perimeter about > 1 m.

Figure 2 caption: Do you mean 'diluvium' (or outwash)?

No, not diluvium. When we wrote "deluvium", we meant loose or poorly cohesive sediment formed as a result of the accumulation of fine mineral particles from soils, clays, till, weathered coverings, etc., washed and eroded from slopes by rainfall and/or mass movements. But we agree it is not appropriate term in English, so to describe this kind of sediments we will change the description to "alluvium and colluvium".

Figure 3: The figure is most illustrative. I suggest, for added clarity, that you specify that the yellow 'V' shapes in panel B and the frost wedge casts.

OK, we will add this explanation to the legend and the figure caption.

Section 3 is very detailed, which is excellent. Such attention to detail can help make your paper a methodologic resource for future work. On line 194, suggest you replace 'decontaminated' with 'isolated', as contamination in the cosmogenic workflow has very different connotations.

Thank you for appreciation of our solid methodical approach and description. We will change the unfortunate word "contaminated" to "isolated".

Section 3.2.1.: Why are you using the Borchers global production rate? The choice of PR is, of course, entirely up to the researchers but given the number of rates now available to use, every choice requires a strong justification in my opinion. This is particularly the case when the Borchers rate contains obviously flawed production 'calibrations' from Scotland. As detailed by Putnam et al. (2019), the inclusion in the global set of production rates calibrated against surfaces of *assumed* (i.e., not actually *known*) age artificially skews that global average rate, making it a little too

high and thus the resulting ages unrealistically young. The CRONUS team themselves noticed the weird impact of the Scottish rates on the overall average (Phillips et al., 2016), pointing out some anomaly in that area, but couldn't (or wouldn't) pinpoint the lack of robust independent dating as the cause. This paper could be improved significantly, therefore, either by using one of the robustly calibrated production rates from the northern mid latitudes (including Europe) or by using the CREp calculator (https://crep.otelo.univ-lorraine.fr/#/), which allows users to remove dubious calibrations from the global primary dataset. I am immediately sceptical of Late Pleistocene studies that blindly use the published Borchers rate, particularly without a very strong justification.

Thank you for this comment/remark. Yes, indeed the production rate for specific cosmogenic nuclide is a critical and very important parameter in surface exposure dating. We used Be-10 production rate from Borchers et al. (2016) as the most recent global production rate in the situation when we do not have any regionally calibrated Be-10 production rate in Poland or generally in Central Europe. Thus, the use of the most recent global data set seems to be the best solution. We note the reviewer's comment about the slight difference between the production rates taking into account or not the Scottish data set in the calibration of the Borchers et al. (2016) Be-10 production rate. The difference between the primary and the secondary production rate calibration data sets from Borchers et al. (2016) will however not change the outcome of our conclusions as the difference is well within the uncertainties linked to the scatter of individual exposure ages. Also using different production rates than these ones based on the global dataset, e.g. Scandinavian reference production rate would not change our ages significantly (Tab. 1). In addition, the exposure ages in our study only provide "background" chronological data as described by the reviewer and as such, are not the pillar of the manuscript in anyway.

Tab. 1. Be-10 data for analyzed samples and surface exposure ages calculated according to various production rates.

| Sample ID | $[^{10}Be]$ $(10^4$ at g$^{-1})$ | Age (ka) | | | |
|---|---|---|---|---|---|
| | | Cronus default production rate (Borchers et al. 2016) | Primary dataset (Borchers et al. 2016) | Secondary dataset (Borchers et al. 2016) | Scandinavian reference production rate (Stroeven et al. 2015) |
| New samples | | | | | |
| LUB-01 | 13.11 ± 0.74 | 25.8 ± 2.4 | 26.0 ± 1.9 | 25.8 ± 2.8 | 26.0 ± 2.1 |
| LUB-02 | 9.87 ± 0.46 | 18.4 ± 1.6 | 18.5 ± 1.2 | 18.4 ± 1.9 | 18.6 ± 1.4 |
| LUB-03 | 7.42 ± 0.47 | 14.1 ± 1.4 | 14.2 ± 1.1 | 14.1 ± 1.6 | 14.3 ± 1.2 |
| LUB-04 | 6.31 ± 0.37 | 12.5 ± 1.2 | 12.6 ± 0.9 | 12.5 ± 1.4 | 12.6 ± 1.0 |
| LUB-05 | 11.58 ± 0.85 | 20.9 ± 2.2 | 21.1 ± 1.8 | 20.9 ± 2.4 | 21.1 ± 2.0 |
| Recalculated samples | | | | | |
| LES-5 | 19.24 ± 1.16 | 40.3 ± 3.9 | 40.6 ± 3.1 | 40.3 ± 4.4 | 40.7 ± 3.4 |
| LES-6 | 8.08 ± 0.58 | 17.4 ± 1.8 | 17.5 ± 1.5 | 17.4 ± 2.0 | 17.5 ± 1.6 |

| | | | | |
|---|---|---|---|---|
| LES-7 | 2.64 ± 0.33 | 5.8 ± 0.8 | 5.8 ± 0.8 | 5.8 ± 0.9 | 5.8 ± 0.8 |
| LES-8 | 10.14 ± 1.10 | 19.7 ± 2.6 | 19.8 ± 2.3 | 19.7 ± 2.8 | 19.9 ± 2.4 |
| LES-10 | 6.78 ± 0.57 | 13.0 ± 1.5 | 13.1 ± 1.2 | 13.0 ± 1.6 | 13.1 ± 1.3 |
| LES-11 | 7.94 ± 0.77 | 16.0 ± 2.0 | 16.1 ± 1.7 | 16.0 ± 2.1 | 16.1 ± 1.8 |
| LES-12 | 8.46 ± 0.70 | 16.1 ± 1.8 | 16.2 ± 1.5 | 16.1 ± 2.0 | 16.2 ± 1.6 |
| LES-13 | 19.15 ± 1.33 | 35.5 ± 3.7 | 35.8 ± 3.0 | 35.5 ± 4.1 | 35.9 ± 3.2 |
| LGM-12 | 11.50 ± 0.53 | 24.1 ± 2.1 | 24.3 ± 1.6 | 24.1 ± 2.5 | 24.4 ± 1.8 |

The differences between the arithmetic mean and the standard deviation for the eleven surface exposure ages analyzed in the manuscript is negligible: **18.0 ± 4.3 ka** using the default production rate dataset set in Cronus, **18.1 ± 4.4 ka** using the primary dataset of Borchers et al. (2016), **18.0 ± 4.3 ka** using the secondary dataset of Borchers et al. (2016) and **18.2 ± 4.4 ka** using the Scandinavian reference production rate (Stroeven et al. 2015).

Reference

Stroeven, A.P., Heyman, J., Fabel, F., Björck, S., Caffee, M.W., Fredin, O., Harbor, J.M., 2015, A new Scandinavian reference 10Be production rate. Quaternary Geochronology 29: 104-115, doi.org/10.1016/j.quageo.2015.06.011.

Line 238: I think the paper needs a clear statement (at some earlier point, not here) about the relevance of an OSL age and the sediments it is dating. My understanding of your stratigraphy is that these are minimum-limiting ages for the RzB2 unit, since the wedges were emplaced into that deglaciated surface. Therefore, the OSL ages do not date the emplacement of the till itself (advance) but the abandonment of that till surface (retreat), and the lag between deglaciation and ice wedge growth/infill cannot be known. Unless I am missing something fundamental here (quite possible), all your OSL ages are minimum ages and thus should be explained as such right from the outset.

Yes, we agree that OSL data are critical for our chronostratigraphy and reconstruction of the timing for ice margin fluctuations in the study area. We will state in the revised version of the manuscript that OSL data are the most relevant and that they are a very solid base for the Bayesian modeling, while beryllium-10 surface-exposure ages give a background for the whole chronology because of their rather large scatter. However, even from this scattered dataset we could note the main mode of Be-10 ages (18.0 ± 4.3 ka) and indicate a possible signal of the ice sheet retreat after ~18 ka, which corresponds to the deglaciation after deposition of till Rz2b and contributed to the overall discussion (lines 317-323).

Yes, of course the stratigraphic relations and the OSL ages of the periglacial sand wedges K1 and K2 give bracketing ages for the possible time for deposition of till units. Wedges K1 must have been formed after deposition of the Rz2a till but before deposition of Rz2b till, and wedge K2 must have been formed after deposition of the Rz2b till but before deposition of Rz2c till. So, OSL ages of K1 wedges are minimum

age for Rz2a till, but maximum age for Rz2b till, and OSL ages of K2 wedge are minimum age for Rz2b till, but maximum age for Rz2c till. They indicate the timing of the periods between till units deposition, and as you noted, not the emplacement of the till itself. For reconstruction of the possible age for the emplacement of the till itself we used Bayesian modelling of the whole sequence – the ages of till layers are modeled ages – Bayesian probability distributions. Although, we tried to describe and explain this in the Methods section (lines 176-185) and in Table A5 in the Appendices, we will state also clearly in the Results section of the revised version of the manuscript which OSL results pre-date and/or post-date the possible timing of the deposition of till units.

Figure 5b: As the authors point out, the Be-10 ages are not very consistent. This is a huge range, particularly in this day and age. Can authors tell us more about why these specific boulders were sampled? Were they all on specific moraine ridges, or are they randomly distributed? In other words, what is their significance? I'm pleased the authors include the beryllium dataset, for it shows complete transparency and no desire to hide 'ugly' data, but a little background would help readers understand better the rationale for sampling and also the potential problems.

Yes, thank you very much for this remark. The reason of sampling these particular boulders was that they are the last big boulders located on moraine ridges (LUB-01, LUB-02, LUB-03 and LUB-05) and not moved/transported by people, available in the study area. So, there was an opportunity to supplement the Be-10 dataset in this region (add new samples to the existing ages from Rinterknecht et al. 2006). The boulder LUB-04 is located in the proglacial valley, however the obtained age (12.5 ± 1.2 ka) is the youngest among the new ages and it suggests that the boulders could be exposed from below the sediments and/or dead ice sometime after deglaciation. The details of all sampled boulders including their geomorphological positions are given in Table A4 in the Appendices. Taking your suggestions into consideration we will give more info about sampled boulders and their geomorphological context in the revised version of the manuscript – we think there is a place for this in section 3.2.

One of the most important reason that probably influenced the scatter of the obtained exposure ages are dynamic geomorphological processes which possibly occured in this area after deglaciation. We stated this briefly in lines 290-291, but in the revised version of the manuscript we will describe/explain these possible processes in a few sentences. Relatively high relief of the study area promotes post-glacial erosional processes, i.e. rainfall washing and/or mass movements along slopes, degradation of the original moraines surface and possible exhumation of erratics from eroded deposits. This could affect the scatter of the obtained ages, despite the fact that almost all of them were selected as boulders resting on moraine crests, in a stable geomorphological position. We may also provide pictures of all sampled boulders as figures in the Appendices.

Lines 276-277: This pattern, should it be correct, is intriguing as it pops up in glacial records worldwide. Net retreat (warming) following the LGM was punctuated by brief pauses or readvances (cooling) that could have existed for just a few years/decades. If the pattern is more widespread than just central Europe, what does that tell us about its climatic drivers and importance? A broader exploration of this possible pattern is warranted, beyond the borders of Poland; if you don't explore it, somebody else will.

Thank you very much for this comment. Indeed, it is very interesting, but we are aware, if such speculations do not push our data interpretation too far... We actually presented an interesting, or as you noted "intriguing", dataset of ages which suggests the millennial-scale fluctuations of the ice margin which can be linked to the offshore marine records. But our results indeed are based on studies within particular region of northern Poland, limited in terms of space, and a broader exploration of this pattern with explanations of the reasons/mechanisms of cooling and warming could be too much and outside the frame of this paper. This kind of exploration/discussion and wider implications you suggested is tempting, but maybe it can be done involving wider dataset/record in a different paper (???).

Line 286: 'Exposition' is not the correct word here.

Yes, we will change to "exposure".

Line 294: I fear the relevance of the OSL ages is being overstepped here. These K1 OSL ages are minimum ages for the till itself, yes? Because the K1 wedges have been emplaced into the till surface *following* deglaciation. You need to make that clear, since the Rz2a till presumably predates the wedges. Yes, you've done bayesian statistics on this age set to get ages for the till, but the truth of the matter is that, from the real data themselves, there is only minimum-limiting age control for the basal till and it could easily postdate 19 ka. Indeed, knowing what we now do about the shape and duration of the LGM, this till could have been emplaced anywhere in MIS-2. Deglaciation subsequently presented a subaerial surface on which periglacial landforms could develop. I'm harping on about this because I think it is important; you could be misrepresenting the age of the basal till by multiple millennia, and that in turn could skew the common perception of when the LGM occurred in these parts (e.g., there is an important difference between, say 24 ka and 19 ka). Already, on Line 338, you are ascribing a concise-sounding age for the basal till (19.1 ± 1.1 ka) that cannot be ascribed based on minimum-limiting OSL ages.

Yes, the stratigraphic sequence and the relations of the periglacial sand wedges to the till layers we explained in details in our above response to the comment to line 238. The possible timing for the deposition of the till layers Rz2a, Rz2b and Rz2c was estimated based on Bayesian modeling, so the ages of tills are modeled ages – Bayesian probability distributions – as also explained in abovementioned response. But these modeled ages were produced based on lithostratigraphic relative relations

and also OSL ages of the whole sediments sequence. We agree that particularly the age (timing of deposition) for the Rz2a till may be debatable, and we noted that in section 5.2 – lines 341-348. We interpreted the age of Rz2a till as 19.2 ± 1.1 ka, based on modeling results. We understand that this is the probability distribution for the most likely age of the till layer, taking into account stratigraphic and OSL constraints. Therefore, we agree that maybe more caution is in order in describing and interpreting these results, and in the revised version of the manuscript we will change an unequivocal statements such as "*The first ice sheet advance which deposited Rz2a till dated at 19.2 ± 1.1 ka...*" into descriptions/explanations that this is the most likely timing for the ice advance according to our modeling results. We will do this both in Results and Discussion sections.

Figure 6: In panels B and D, how do you know the ice margin retreated outside the study area like this? What is your geologic evidence for that? I don't see any described in this manuscript. Likewise, in panels C and E, what is your geologic evidence for the ice margin having stabilised at these tidy blue lines? Are there conspicuous moraine complexes defining a robust, stable ice margin, or is this conjecture? If the latter, please specify and use dashed lines rather than filled; otherwise, folk might take this as true when it could be little more than speculation. I assume the glacial geology of these parts has been thoroughly mapped?

Thank you for these comments, it is very important to clarify these issues. So for situation depicted in panels B and D, we actually do not have direct evidences that the ice sheet retreated beyond the study area at these stages. The only thing which we know is that the ice did not cover the Rożental locality at these times (because of periglacial sand wedges). So the ice margin theoretically could be located north-east of the Rożental site, but still within the study area, or beyond the study area. In the revised version of the manuscript we will modify the statement such as "*The ice margin retreated to the north and north-west of the study area and periglacial conditions occurred again...*" and describe that the margin could be located beyond the study area or within the north-eastern corner of the study area, and that the periglacial conditions occurred at least in the most part of the study area. We will also modify the maps in panels B and D – we will insert ice margins with dashed lines and question marks.

For the evidences that the ice margin stabilized along the blue lines in panels C and E – we mentioned some geomorphological evidences in the Discussion (lines 313-316 and 327-328). However, probably it is reasonable to show them in detail. We enclosed here a series of closer looks for the landform and sediments distribution and our interpretation of the ice limits. They are referred to panels C and E in Fig. 6:

[Figure]

Fig. 1. A series of closer looks for the interpreted positions of the ice margin during particular ice re-advances. Landforms and sediments distribution across the study area is based on Detailed Geological Map of Poland (Gałązka and Marks, 1997; Gałązka, 2003, 2006, 2009; Wełniak, 2002).

Line 316: Yet, this is essentially speculation. Again, please be careful, as Figure 6 will give the impression that these speculative margins are based on unequivocal geologic mapping - and they aren't. Again, I suggest you use dashed lines instead and state specifically in the caption that these are entirely speculative.

As we mentioned and showed above, the ice sheet limits are interpreted based on spatial distribution of landforms and sediments, namely: outlets of tunnel channels, ice-marginal channels, end moraines as well as proximal edges of outwash plains. This is indicated also in the text (lines 313-316 and 327-328), so the ice margin positions are not entirely speculative. But of course we understand it is a matter of our interpretation for the ice margin position, so we may mark ice-sheet limits into the dashed lines in Fig. 6 and describe/explain that these are the probable ice-margin stillstand inferred from spatial distribution of landforms and sediments.

Lines 327-329: Again, there's a lot of speculation here. Why not include some detailed mapping? That would make this a much stronger contribution.

Yes, we provided a figure with details regarding the interpretation of the ice margin positions above. In the revised version of the manuscript we may also add a new figure with closer look on digital elevation model and geologic map, which will show the arguments supporting our view on ice margin configuration presented on panels C and E in Fig. 6 (???).

Line 348: Again, this should be defined as a minimum-limiting age.

We have already referred to this issue above.

Line 385: I understand the logic here, but this relationship is again highly speculative. What is the global nature of this inferred readvance? How is it possible to correlate with a Heinrich event if we don't, as a community, yet know quite what causes H events? (Or why some stadials include them and others don't, or some H events occur without stadials?) And when accurate and precise C-14 dating of H events in marine records continues to elude us due to reservoir uncertainties? I think this is saying way too much given both the uncertainties in your data (excellent OSL ages, but they do have sizeable error bars, as is expected) and the persistent lack of understanding regarding H events themselves. I'm not saying don't hypothesise, rather I just urge more caution.

Yes, thank you for this comment. What we did in this part of the Discussion is to correlate our ice margin fluctuations with Heinrich events and marine record. We agree it is speculative, but this speculation is based on data and chronology we obtained. We understand and are aware that our results are based on studies within particular region of northern Poland, limited in terms of space (as we stated above), so that there is still much work to do to confirm/explore further this interesting pattern. However, we also argue that this case study is the first (?) terrestrial record corresponding to the millennial scale fluctuations recorded offshore, which may be very interesting and inspiring for the community. On the other hand, we again agree that maybe more caution is necessary when interpreting these results, and in the revised version of the manuscript we may change the statement such as "*After ~17.5–17.0 ka the meltwater discharge from the southern FIS significantly decreased in*

*response to the initiation of a deglacial pause and a global re-advance of glaciers and ice sheets in Europe corresponding to Heinrich event 1 (HE1) (Zaragossi et al., 2001; Toucanne et al., 2009). This event is correlated to the last ice advance recorded in our study area and dated at 16.9 ± 0.5 ka (Fig. 6f)."* (lines 383-386) into descriptions/explanations that this is likely and that we hypothesize this, but unequivocal correlation goes maybe too far.

References cited:

Phillips, F.M., Argento, D.C., Balco, G., Caffee, M.W., Clem, J., Dunai, T.J., Finkel, R., Goehring, B., Gosse, J.C., Hudson, A.M. and Jull, A.T., 2016. The CRONUS-Earth project: a synthesis. Quaternary Geochronology, 31, pp.119-154.

Putnam, A.E., Bromley, G.R., Rademaker, K. and Schaefer, J.M., 2019. In situ 10Be production-rate calibration from a 14C-dated late-glacial moraine belt in Rannoch Moor, central Scottish Highlands. Quaternary Geochronology, 50, pp.109-125.

---

## Author Comment (AC2)

We differentiated with colors our responses for the reviewers comments and questions. While the reviewers comments and questions are indicated in black, our responses are written in green. Response for particular comment immediately below the comment.

**RC2**: 'Comment on tc-2023-117', Anonymous Referee #2, 05 Nov 2023

This manuscript provides new information on Late Pleistocene ice margin fluctuations at the southern fringe of the Fennoscandian Ice Sheet. The new evidence comprises 11 OSL measurements from fluvioglacial and periglacial deposits exposed in the Rożental gravel pit, and 5 $^{10}$Be boulder surface exposure ages, all derived from a 30km$^2$ area of the moraine plateau in northern Poland. Unfortunately, the $^{10}$Be ages do not contribute to the discussion, however it is important that they were presented here for sake of transparency.

Yes of course, OSL data are the most important in our reconstruction, but we do not agree that the Be-10 ages do not contribute to the discussion. Even from this scattered dataset we could note the main mode of Be-10 ages (18.0 ± 4.3 ka) and indicate a possible signal of the ice sheet retreat after ~18 ka, what corresponds to the deglaciation after deposition of till Rz2b and contributed to the overall discussion (lines 317-323).

The millennial scale fluctuations are based on the Bayesian modelling of the OSL data from Rożental gravel pit, where 3 glacial tills separate periglacial sedimentary features. The authors link their millennial scale fluctuations to evidence of meltwater sediment deposition in the Bay of Biscay. This apparent relationship is speculative but interesting. Overall, the manuscripts presents new terrestrial data on ice margin fluctuations in northern Poland and is well within the scope of TC. The paper reaches substantial conclusions which are backed up by the evidence presented. The methodology is well described and all relevant data necessary for readers to recalculate the ages is provided. References are appropriate, but some additional ones are suggested for Figure 1. The supplementary materials are generally good, boulder sizes are missing in Table A4.

Thank you for your opinion. Yes, we agree that some of our interpretations/discussion are speculative, but we tried to speculate based on the results we obtained. Our opinion regarding this was also stated in the responses for the Referee #1, in particular for the comments regarding lines 276-277 and 385.

Below I have attempted to correct the grammar and also make more detailed comments. LXX refers to Line XX in the submitted manuscript.

Thank you very much for all grammar/vocabulary/style corrections, we will correct the manuscript according to the all below suggestions.

Suggested edits:

L15: Optically stimulated luminescence (OSL) was used to date…

L17: 10Be surface exposure dating was used…

L18: …resting on the surface…

L23: "confront" is not the right word, compare is better

L35: The sentence "The last two ice sheets….Rignot et al., 2019)." can be removed. It does not add anything meaningful.

L40: geological records

L41: enables correlating

L43: glacial records…nature of the glacial

L48: see comment for L15, or use OSL

L49: see comment for Line17

L52: see comment for L23

L59: …within an elevated…

L62: denivelations should be relief

L64: sheet's should be sheet

L65: The sediment outcrop where…

L78: …of a well-preserved…delete "a"

L86: …and it consists of…should be …and consist of…

L109: …most of surface… should be …most surface…

L110: massive is subjective, give dimensions

L110-111: Suggest changing ", which commonly occur at the surface of moraine plateau,…" to "located on the moraine plateau,…"

L114: deluvium, or diluvium, is an obsolete term. Is this alluvium?

Yes, it is a wrong word in English.  We mean loose or poorly cohesive sediment formed as a result of the accumulation of fine mineral particles from soils, clays, till, weathered coverings, etc., washed and eroded from slopes by rainfall and/or mass movements. To describe this kind of sediments we will change the description to "alluvium and colluvium".

L120: The sequence…

L122: "reveals" should be "show"

L125: "with a fossil" should be "with fossil"

L142: PCV should be PVC

L147: "remaining of other" should be "remaining other"

L157: "decay chain and potassium" should be "decay chains of potassium"

L158: delete "the reference materials, namely"

L164: "consists of sequence of sediments units" should be "consists of the sequence of sediment units"

L166: "sediments deposition" should be "sediment deposition"

L166: *likehood*  should be *likelihood* make sure you check every occurrence.

L172: "sediments deposition" should be "sediment deposition"

L175: "pre-date particular event" should be "pre-date a particular event"

L184: "with *IntCal20*" should be "with the *IntCal20*"

L187: see comment for L110

L190: "lithologies as" should be "lithologies such as"

L194: "decontaminated" should be "separated"

L232: "samples MAM" should be "samples the MAM"

L133: "as Rz1" should be "as the Rz1"

L234: "It is visible especially within aliquots distribution of…" should be "This is especially visible in the aliquot distributions of…"

L236: "deposition of Rz1 unit" should be "deposition of the Rz1 unit"

L241: "dominates" should be "dominate"

L242: "revealing" should be "shows a", "with CAM" should be "with the CAM"

L248: "with CAM" should be "with the CAM"

L286: What do you mean by boulder exposition after deglaciation and/or significant postglacial erosion… How much erosion would you need to explain the young age? Why would this boulder erode so much faster than the others you sampled? Please provide a reasoned argument for your explanations of the young age. The most likely reason for the young age is that the boulder has moved or that it was buried and subsequently exposed. Could this be the case?

Yes, agree. Probably some boulders were buried and subsequently exposed. Relatively high relief of the study area promotes post-glacial erosional processes, i.e. rainfall washing and/or mass movements along slopes, degradation of the original moraines surface and possible exhumation of erratics from eroded deposits. We will correct the explanation of the young age accordingly in the revised version of the manuscript.

L320: "of boulder" should be "of a boulder"

L341: "for Rz1" should be "for the Rz1"

L342: "of Rz2a" should be "of the Rz2a"

L349: "formation" should be "formations"

L350: "in north-" should be "in the north-"

L351: "be also related to regional" should be "also be related to a regional"

L354: "a large" delete "a"

L361: ~17 ka and" delete "and"

L366: "record" should be "records"

L368: "been already" should be "already been"

L376: "sediments" should be "sediment"

L377: "sediments" should be "sediment"

L403: "record" should be "records"

Figure 1 caption: provide references for the LGM and PM ice margins. The outline script for the LUB samples and Rożental reduces clarity. Either use solid white or increase the size of the labels. The Gd coring site label in the legend does not match the style on the map. Label the grey dashed box as Figure 2.

OK, we will correct the figure according to your suggestion.

Figure 2 caption: What is the grey dashed box? What is deluvium? If you mean diluvium, it is an obsolete term. Is this material fluvial or glacial? Please check appropriate terminology.

The grey dashed box is the area of the figure 5b. We should make the label more visible. Regarding "deluvium" – we explained above.

Figure 5 caption: "(a) Sediments profile" should be "(a) Sediment profile"

Figure 6: Drawing ice limits on maps is one of the hardest processes to justify to the community. The process probably generated quite a lot of discussion and various versions of the maps. What are the uncertainties of the isochrons on the map, and how were the locations of these limits established. The explanation in the text suggests ambiguity "not equivocal" about the exact position. Could this ambiguity be shown on the map with a maximum/minimum limit?

Yes, thanks for this remark. The question of the ice-margin positions showed in Fig. 6 was also asked by Referee #1. We explained as far as we could the basis for interpreting ice-margin positions on panels C and E in Fig. 6 and gave some closer looks on digital elevation model with particular landforms suggesting ice-margin positions in our response for Referee #1. We will mark the ice margin positions as "possible" positions inferred from landforms and sediments spatial distribution in the study area, and with dashed lines rather than solid in the revised version of the manuscript. We hope this will show the ambiguity "not equivocal" in this case.

---

## Author Response (AR1)

We differentiated with colors our responses for the reviewers comments and questions. While the reviewers comments and questions are indicated in black, our responses are written in green. Response for particular comment immediately below the comment.

We revised the manuscript according to reviewers suggestions and remarks. All changes made in the manuscript text are visible in the track-changes file uploaded together with revised manuscript. Figures were also changed and corrected according to reviewer's comments.

**Anonymous Referee #1**

Abstract: Look out for typographical errors. This is a fine introduction/synthesis of the manuscript but the incomplete English distracts the reader's attention somewhat. The same goes for the rest of the manuscript, which is well written but would benefit from fine tuning by a native English speaker.

We tried to correct the typographical errors and the general language quality of the manuscript. Referee #2 pointed out a several language suggestions and remarks to the particular statements/words in the text. We corrected them all.

Figure 1: Legend is a little confusing. The black circles with values are recalculated ages, yes? What are the black circles without values? It would also strengthen the figure to justify the positions of the LGM and Pomeranian ice margins – whose work confirms that these positions are correct? What about chronology? Without those details it looks a little like guesswork.

We corrected the figure caption with appropriate explanation. We also added reference related to the lines indicating maximum extent and Pomeranian Phase (Marks, et al., 2006).

Line 110: Such things are subjective. How big is massive? Give approx. size.

We specified that we are talking about boulders with perimeter about ≥ 1 m (lines 122 and 199).

Figure 2 caption: Do you mean 'diluvium' (or outwash)?

We changed to "alluvium and colluvium".

Figure 3: The figure is most illustrative. I suggest, for added clarity, that you specify that the yellow 'V' shapes in panel B and the frost wedge casts.

We added a proper explanation in the figure caption.

Section 3 is very detailed, which is excellent. Such attention to detail can help make your paper a methodologic resource for future work. On line 194, suggest you replace 'decontaminated' with 'isolated', as contamination in the cosmogenic workflow has very different connotations.

We changed to "separated" (line 210).

Section 3.2.1.: Why are you using the Borchers global production rate? The choice of PR is, of course, entirely up to the researchers but given the number of rates now available to use, every choice requires a strong justification in my opinion. This is particularly the case when the Borchers rate contains obviously flawed production 'calibrations' from Scotland. As detailed by Putnam et al. (2019), the inclusion in the global set of production rates calibrated against surfaces of *assumed* (i.e., not actually *known*) age artificially skews that global average rate, making it a little too high and thus the resulting ages unrealistically young. The CRONUS team themselves noticed the weird impact of the Scottish rates on the overall average (Phillips et al., 2016), pointing out some anomaly in that area, but couldn't (or wouldn't) pinpoint the lack of robust independent dating as the cause. This paper could be improved significantly, therefore, either by using one of the robustly calibrated production rates from the northern mid latitudes (including Europe) or by using the CREp calculator (https://crep.otelo.univ-lorraine.fr/#/), which allows users to remove dubious calibrations from the global primary dataset. I am immediately sceptical of Late Pleistocene studies that blindly use the published Borchers rate, particularly without a very strong justification.

We used Be-10 production rate from Borchers et al. (2016) as the most recent global production rate in the situation when we do not have any regionally calibrated Be-10 production rate in Poland or generally in Central Europe. Thus, the use of the most recent global data set seems to be the best solution. We note the reviewer's comment about the slight difference between the production rates taking into account or not the Scottish data set in the calibration of the Borchers et al. (2016) Be-10 production rate. The difference between the primary and the secondary production rate calibration data sets from Borchers et al. (2016) will however not change the outcome of our conclusions as the difference is well within the uncertainties linked to the scatter of individual exposure ages. Also using different production rates than these ones based on the global dataset, e.g. Scandinavian reference production rate would not change our ages significantly (Tab. 1). In addition, the exposure ages in our study only provide "background" chronological data as described by the reviewer and as such, are not the pillar of the manuscript in anyway.

Tab. 1. Be-10 data for analyzed samples and surface exposure ages calculated according to various production rates.

| Sample ID | [¹⁰Be] (10⁴ at g⁻¹) | Age (ka) | | | |
|---|---|---|---|---|---|
| | | Cronus default production rate (Borchers et al. 2016) | Primary dataset (Borchers et al. 2016) | Secondary dataset (Borchers et al. 2016) | Scandinavian reference production rate (Stroeven et al. 2015) |
| New samples | | | | | |
| LUB-01 | 13.11 ± 0.74 | 25.8 ± 2.4 | 26.0 ± 1.9 | 25.8 ± 2.8 | 26.0 ± 2.1 |
| LUB-02 | 9.87 ± 0.46 | 18.4 ± 1.6 | 18.5 ± 1.2 | 18.4 ± 1.9 | 18.6 ± 1.4 |
| LUB-03 | 7.42 ± 0.47 | 14.1 ± 1.4 | 14.2 ± 1.1 | 14.1 ± 1.6 | 14.3 ± 1.2 |
| LUB-04 | 6.31 ± 0.37 | 12.5 ± 1.2 | 12.6 ± 0.9 | 12.5 ± 1.4 | 12.6 ± 1.0 |
| LUB-05 | 11.58 ± 0.85 | 20.9 ± 2.2 | 21.1 ± 1.8 | 20.9 ± 2.4 | 21.1 ± 2.0 |
| Recalculated samples | | | | | |
| LES-5 | 19.24 ± 1.16 | 40.3 ± 3.9 | 40.6 ± 3.1 | 40.3 ± 4.4 | 40.7 ± 3.4 |
| LES-6 | 8.08 ± 0.58 | 17.4 ± 1.8 | 17.5 ± 1.5 | 17.4 ± 2.0 | 17.5 ± 1.6 |
| LES-7 | 2.64 ± 0.33 | 5.8 ± 0.8 | 5.8 ± 0.8 | 5.8 ± 0.9 | 5.8 ± 0.8 |
| LES-8 | 10.14 ± 1.10 | 19.7 ± 2.6 | 19.8 ± 2.3 | 19.7 ± 2.8 | 19.9 ± 2.4 |
| LES-10 | 6.78 ± 0.57 | 13.0 ± 1.5 | 13.1 ± 1.2 | 13.0 ± 1.6 | 13.1 ± 1.3 |
| LES-11 | 7.94 ± 0.77 | 16.0 ± 2.0 | 16.1 ± 1.7 | 16.0 ± 2.1 | 16.1 ± 1.8 |
| LES-12 | 8.46 ± 0.70 | 16.1 ± 1.8 | 16.2 ± 1.5 | 16.1 ± 2.0 | 16.2 ± 1.6 |
| LES-13 | 19.15 ± 1.33 | 35.5 ± 3.7 | 35.8 ± 3.0 | 35.5 ± 4.1 | 35.9 ± 3.2 |
| LGM-12 | 11.50 ± 0.53 | 24.1 ± 2.1 | 24.3 ± 1.6 | 24.1 ± 2.5 | 24.4 ± 1.8 |

The differences between the arithmetic mean and the standard deviation for the eleven surface exposure ages analyzed in the manuscript is negligible: **18.0 ± 4.3 ka** using the default production rate dataset set in Cronus, **18.1 ± 4.4 ka** using the primary dataset of Borchers et al. (2016), **18.0 ± 4.3 ka** using the secondary dataset of Borchers et al. (2016) and **18.2 ± 4.4 ka** using the Scandinavian reference production rate (Stroeven et al. 2015).

Reference

Stroeven, A.P., Heyman, J., Fabel, F., Björck, S., Caffee, M.W., Fredin, O., Harbor, J.M., 2015, A new Scandinavian reference 10Be production rate. Quaternary Geochronology 29: 104-115, doi.org/10.1016/j.quageo.2015.06.011.

Line 238: I think the paper needs a clear statement (at some earlier point, not here) about the relevance of an OSL age and the sediments it is dating. My understanding of your stratigraphy is that these are minimum-limiting ages for the RzB2 unit, since the wedges were emplaced into that deglaciated surface. Therefore, the OSL ages do not date the emplacement of the till itself (advance) but the abandonment of that till surface (retreat), and the lag between deglaciation and ice wedge growth/infill cannot be known. Unless I am missing something fundamental here (quite possible), all your OSL ages are minimum ages and thus should be explained as such right from the outset.

We stated in the revised version of the manuscript that OSL data are the most relevant and that they are a very solid base for the Bayesian modeling (lines 20, 291-293, 421-422).

We stated clearly in the Results section of the revised manuscript that horizon K1 must has been formed after deposition of the Rz2a till and before the Rz2b till, and that horizon K2 must has been formed after deposition of the Rz2b till and before Rz2c (lines 257-259 and 264-265).

Figure 5b: As the authors point out, the Be-10 ages are not very consistent. This is a huge range, particularly in this day and age. Can authors tell us more about why these specific boulders were sampled? Were they all on specific moraine ridges, or are they randomly distributed? In other words, what is their significance? I'm pleased the authors include the beryllium dataset, for it shows complete transparency and no desire to hide 'ugly' data, but a little background would help readers understand better the rationale for sampling and also the potential problems.

We gave more info about sampled boulders and their geomorphological context in the revised version of the manuscript, in section 3.2 (lines 200-205).

We explained that "*relatively high relief of the study area promotes post-glacial erosional processes, i.e. rainfall washing and/or mass movements along slopes, degradation of the original moraines surface and possible exhumation of erratics from eroded deposits*" in the Results section (lines 302-305). We also provided additional figures with detailed geomorphological location of sampled boulders in the Appendices.

Lines 276-277: This pattern, should it be correct, is intriguing as it pops up in glacial records worldwide. Net retreat (warming) following the LGM was punctuated by brief pauses or readvances (cooling) that could have existed for just a few years/decades. If the pattern is more widespread than just central Europe, what does that tell us about its climatic drivers and importance? A broader exploration of this possible pattern is warranted, beyond the borders of Poland; if you don't explore it, somebody else will.

We argue that a broader exploration of this pattern with explanations of the reasons/mechanisms of cooling and warming could be too much and outside the frame of this paper. This kind of exploration/discussion and wider implications you suggested is tempting, but maybe it can be done involving wider dataset/record in a different paper (???).

Line 286: 'Exposition' is not the correct word here.

We changed to "exposure".

Line 294: I fear the relevance of the OSL ages is being overstepped here. These K1 OSL ages are minimum ages for the till itself, yes? Because the K1 wedges have been emplaced into the till surface *following* deglaciation. You need to make that clear, since the Rz2a till presumably predates the wedges. Yes, you've done bayesian statistics on this age set to get ages for the till, but the truth of the matter is that, from the real data themselves, there is only minimum-limiting age control for the basal till and it could easily postdate 19 ka. Indeed, knowing what we now do about the shape and duration of the LGM, this till could have been emplaced anywhere in MIS-2. Deglaciation subsequently presented a subaerial surface on which periglacial landforms could develop. I'm harping on about this because I think it is important; you could be misrepresenting the age of the basal till by multiple millennia, and that in turn could skew the common perception of when the LGM occurred in these parts (e.g., there is an important difference between, say 24 ka and 19 ka). Already, on Line 338, you are ascribing a concise-sounding age for the basal till (19.1 ± 1.1 ka) that cannot be ascribed based on minimum-limiting OSL ages.

The possible timing for the deposition of the till layers Rz2a, Rz2b and Rz2c was estimated based on Bayesian modeling, so the ages of tills are modeled ages – Bayesian probability distributions based on lithostratigraphic relative relations and OSL ages of the whole sediments sequence – we tried to emphasize this in the revised manuscript. We also changed an unequivocal statements in Results and discussion sections such as "*The first ice sheet advance which deposited Rz2a till dated at 19.2 ± 1.1 ka...*" into descriptions/explanations that this is the most likely timing for the ice advance according to our modeling results (lines 289-293, 357, 368, 376, 421-424).

Figure 6: In panels B and D, how do you know the ice margin retreated outside the study area like this? What is your geologic evidence for that? I don't see any described in this manuscript. Likewise, in panels C and E, what is your geologic evidence for the ice margin having stabilised at these tidy blue lines? Are there conspicuous moraine complexes defining a robust, stable ice margin, or is this conjecture? If the latter, please specify and use dashed lines rather than filled; otherwise, folk might take this as true when it could be little more than speculation. I assume the glacial geology of these parts has been thoroughly mapped?

We modified the maps in panels B and D – now there is ice sheet, but with question marks. Also in the text we stated that the ice sheet retreated to NW of the Rożental site, but we do not know for sure if these were retreats beyond the study area (lines 335-336).

Line 316: Yet, this is essentially speculation. Again, please be careful, as Figure 6 will give the impression that these speculative margins are based on unequivocal geologic mapping - and they aren't. Again, I suggest you use dashed lines instead and state specifically in the caption that these are entirely speculative.

We changed the marked ice-sheet limits into the dashed lines in Fig. 6 and describe/explain that these are the probable ice-margin positions.

Lines 327-329: Again, there's a lot of speculation here. Why not include some detailed mapping? That would make this a much stronger contribution.

We provided a figure with details regarding the interpretation of the ice margin positions (DEM and surface deposits) in our response for the review in the interactive discussion part.

Line 348: Again, this should be defined as a minimum-limiting age.

We stated in the revised manuscript what is the relation of periglacial horizons to till layers (lines 257-259 and 264-265).

Line 385: I understand the logic here, but this relationship is again highly speculative. What is the global nature of this inferred readvance? How is it possible to correlate with a Heinrich event if we don't, as a community, yet know quite what causes H events? (Or why some stadials include them and others don't, or some H events occur without stadials?) And when accurate and precise C-14 dating of H events in marine records continues to elude us due to reservoir uncertainties? I think this is saying way too much given both the uncertainties in your data (excellent OSL ages, but they do have sizeable error bars, as is expected) and the persistent lack of understanding regarding H events themselves. I'm not saying don't hypothesise, rather I just urge more caution.

Yes, we tried to change this part of the Discussion in the revised manuscript to emphasize, that such correlation is probable, that our result suggest this, but that it is not unequivocal (lines 402-406).

References cited:

Phillips, F.M., Argento, D.C., Balco, G., Caffee, M.W., Clem, J., Dunai, T.J., Finkel, R., Goehring, B., Gosse, J.C., Hudson, A.M. and Jull, A.T., 2016. The CRONUS-Earth project: a synthesis. Quaternary Geochronology, 31, pp.119-154.

Putnam, A.E., Bromley, G.R., Rademaker, K. and Schaefer, J.M., 2019. In situ 10Be production-rate calibration from a 14C-dated late-glacial moraine belt in Rannoch Moor, central Scottish Highlands. Quaternary Geochronology, 50, pp.109-125.

---

## Author Response (AR2)

Dear authors,

Thanks for submitting a revised version of your manuscript. It addresses most of the concerns of the reviewer. However, I still feel that you should include the answer you gave to reviewer 1's comment about the Borcher's global production rate within your manuscript, more specifically within section 3.2.1, or at the end of section 5.1 and include the table you provide in your answer to reviewer in the supplementary material. I think this is important to include all elements for discussion so that one can have a proper critical analysis of the importance of the outcomes of your research. In addition, including those elements will also strengthen the conclusions of this manuscript. After doing this, your manuscript will be ready for publication.

Dear Editor,

Thank you very much for your remark about the information regarding $^{10}$Be production rates, which should be included in the manuscript. We fully agree it will strengthen the conclusions of this manuscript. According to your suggestions we included appropriate information in the section 3.2.1 (lines 226-231) and also in the section 5.1 (lines 335-340 in the revised version of the manuscript). As a consequence, there is one additional reference (Stroeven et al., 2015) added to the reference list in the revised version of the manuscript. We also added a table with the comparison of exposure ages calculated with various $^{10}$Be production rates to the Appendices (Table A5). We hope this will make the manuscript suitable for publication.